# mmSSR: Harvesting Rich, Scalable and Transferable Multi-Modal Data for Instruction Fine-Tuning

## Abstract

The hypothesis that pretrained large language models (LLMs) necessitate only minimal supervision during the fine-tuning (SFT) stage has been substantiated by recent advancements in data curation and selection research. However, their stability and generalizability are compromised due to the vulnerability to experimental setups and validation protocols, falling short of surpassing random sampling. Built upon LLMs, multi-modal LLMs (MLLMs), combined with the sheer token volume and heightened heterogeneity of data sources, amplify both the significance and complexity of data selection. To harvest multi-modal instructional data in a robust, efficient and transferable manner, we re-define the granularity of the quality metric by decomposing it into more than ten vision-language-related interpretable capabilities, and introduce multi-modal rich scorers to evaluate the corresponding value for each sample. In light of the inherent objective of the instructional stage, we take interactive styles as a superficial diversity indicator, and use a multi-modal rich styler to partition candidate data. In doing so, our **m**ulti-**m**odal **r**ich **s**corers and **s**tyler (mmSSR) guarantee that high-scoring information is delivered to users in diversified forms. Free from embedding-based clustering or greedy sampling, mmSSR efficiently scales to millions of data with varying budget constraints, supports general and specific capability customization, and facilitates training-free transfer to new domains for curation. Across 10+ experimental settings, validated by 14 multi-modal benchmarks, we demonstrate consistent improvements over random sampling, baseline strategies and state-of-the-art selection methods, achieving 99.1% of full performance using only 30% of the 2.6M data.

## 1 Introduction

The quality of data matters in the scaling of large models (Li et al., 2024b; Wettig et al., 2024; Liu et al., 2024b; Lu et al., 2024; Luo et al., 2024; Li et al., 2024a). It is particularly important during their supervised fine-tuning (SFT) stage, where pre-trained models are expected to efficiently and accurately follow user instructions for general purposes or specialized deployment. To achieve this, earlier approaches for large language models (LLMs) filter large-scale SFT datasets with millions of samples towards redundancy reduction (Lee et al., 2022; Elazar et al., 2024), quality control and safety regulation (Joulin, 2016; Penedo et al., 2023; Dubey et al., 2024; Team et al., 2024; Chung et al., 2024). Recently, LIMA introduces the superficial alignment hypothesis (SAH) (Zhou et al., 2024), which utilizes only 1,000 carefully curated samples to illustrate that most LLM knowledge has been acquired during pre-training, requiring only minimal data for instruction fine-tuning, and the effectiveness of these few samples hinges on their quality and diversity. This shift has encouraged subsequent research on automated sample selection, which aims to identify and extract valuable data on these key attributes (Lu et al., 2024; Xia et al., 2024a; Liu et al., 2025), thereby reducing time and computational cost while enhancing interpretability of the target models. However, although the SAH remains valid under the verification of hand-crafted data, recent surveys (Diddee & Ippolito, 2024; Xia et al., 2024b) reveal that automated sample selection methods are susceptible to experimental conditions, including variations in available budgets, different data sources and diverse evaluation benchmarks, which hinders them to get consensus on benchmarks or consistently outperform uniform sampling in generalization. And their reliance on data embedding to promote subset diversity could

end up making the entire process inefficient and unable to scale up (Liu et al., 2024b; Pang et al., 2024; Li et al., 2024c).

Building on the challenges in data selection for LLMs, we shift our focus to multi-modal LLMs (MLLMs) with *millions of finetuning data*, where the increased variety of data modalities, combined with the sheer token volume and heterogeneity of data sources, elevate the significance of data selection as a critical yet underexplored aspect of model performance. First, unlike their text-only counterparts, the selection algorithm must be adept at identifying samples that not only exhibit high quality and diversity within each modality but capture the underlying correlations between them. On the other hand, MLLMs pose new challenges in achieving generalization across various settings and tasks due to the pronounced noise and variability inherent in the multi-modal data curation process (Chen et al., 2024a; Li et al., 2024a; Liu et al., 2024a). Furthermore, the sensitivity of sample selection methods of LLMs prevents their direct adaptation to MLLMs, and the vision-language (VL) alignment metrics adopted by VL models (VLMs) (Maini et al., 2024; Gadre et al., 2023) is not aligned with the motivation of instruction alignment, showing suboptimal performance (See our results in Sec. 4.2) These observations necessitate innovative approaches for mutli-modal data selection to cut computational consumption and improve data understanding (Wang et al., 2024a).

In response to the challenges of multi-modal data selection valuation in coverage, data scaling and transferability, we propose to decompose the complexity of data into rich capabilities that are human-interpretable and model-attributable (such as spatial understanding, logical deduction), which breaks down abstract concepts into multiple concrete metrics that can be systematically evaluated. In this paper, we exemplify more than ten criteria that serve as the foundational pillars

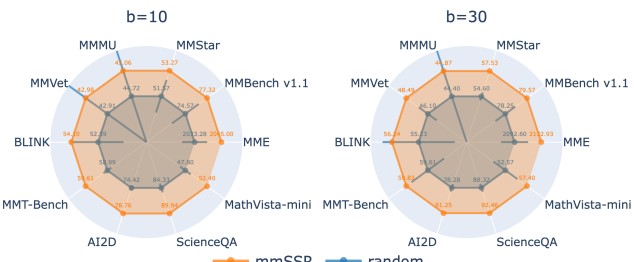

Figure 1: mmSSR against the random sampling baseline across both general and specialized multi-modal benchmarks under the 10% and 30% data budgets.

for the development of vision perception and reasoning capabilities, and train corresponding scorers to provide assessments on each candidate. In comparison to vague formulation such as *quality* or *complexity* (Liu et al., 2024b; Pang et al., 2024), our rich scores re-define the granularity of data valuation, facilitating improved understanding, easy customization and better transferability. Different from potentially task-specific metric or model-dependent predictions, the concrete criteria we propose carry clear and general semantics that can be easily exposed from the pre-trained model, so that our instructed scorers will not overfit to the training data if the existing data pool is limited, yielding robust generalization capabilities across tasks and domains. It also shows significant improvements in efficiency and practicality in comparison to influence estimation methods (Wu et al., 2024), which necessitates access to both training and validation sets. Equipped with rich scores of multi-modal instances, ensuring data diversity becomes a critical next step, especially for large-scale multi-modal heterogeneous mixtures. In light of the nature of the instruction tuning stage, where the model learns to interact with users in different styles (Zhou et al., 2024), we take the superficial instruction styles as a straightforward indicator of diversity, and introduce a multi-modal rich styler to cluster instances based on their interaction patterns. Free from in-domain feature representation learning (Lee et al., 2024; Wu et al., 2024), distance-based greedy filtering, cluster-based sampling (Liu et al., 2024b; Lee et al., 2024), the instance-level style clustering significantly reduces computational complexity and becomes scalable. In our experiments on the LLaVA-OneVision (LLaVA-OV) (Li et al., 2024a), the state-of-the-art (SOTA) open MLLM series with a well-curated dataset, we demonstrate the significance of our **m**ulti-**m**odal **R**ich **S**corer and **S**tyler (mmSSR) across 6 varying budget settings and 2 different model sizes, comprehensively validated with 14 benchmarks. We further evaluate the practicality of mmSSR towards domain generalization and its scalability in data quantity and capability, which demonstrates efficient adaptation, flexible customization and potential for data scaling. The main contributions are summarized as follows:

- We present a novel data selection pipeline for multi-modal instruction data, which decomposes the task complexity into rich capabilities and styles for data valuation and diversity.
- mmSSR demonstrates superiority in performance, scalability and transferability, as comprehensively validated across 10 settings with 14 general and specialized benchmarks.

- The pre-tuned mmSSR, along with our scoring data and selected subsets, can be readily utilized by the community for domain generalization and new capability acquisition.

## 2    RELATED WORK

Recent advances have explored various strategies to improve data efficiency. While research on MLLMs remains limited, our work draws inspiration from existing studies for LLMs, vision-language models (VLMs), and active learning. These efforts can be broadly categorized hand-crafted heuristics, model-based indicators, and LLM-based scoring. They can be inter-changeably or complementarily applied across different training stages.

**Hand-Crafted Heuristics**    rely on expert knowledge on the specific task to establish quality metrics for data filtering or selection. From high-level features such as relevance, clarity, diversity and safety to lower-level indicators like vocabulary, N-gram and sentence length (Team et al., 2023; Touvron et al., 2023; Dubey et al., 2024; Qin et al., 2024; Penedo et al., 2023). While these heuristics are interpretable and straightforward to implement, they are labor-intensive and often prone to human bias, and lack adaptability to iterate target models and to multi-modal data challenges.

**Model-Based Indicators**    often leverage the internal mechanisms or outputs of target models to assess data. Across machine learning algorithms and recent large models, a common paradigm leverages the gradients, predictive distributions, and embeddings of the target model to assess uncertainty, entropy, learnability, similarity and transferability (Evans et al., 2024; Liu et al., 2024c; Lyu et al., 2023; 2025; Sener & Savarese, 2018; Liu et al., 2024b; 2025; Settles, 2009). These approaches could offer promising in-domain performance when the computational cost of target models are affordable. However, judgments made by models may also struggle with interpretability and transferability (Diddee & Ippolito, 2024; Munjal et al., 2020). Introducing proxy models into the data selection pipeline mitigates dependency on the task model. One widely adopted strategy is to train bigram or unigram classifiers (Joulin, 2016; Brown et al., 2020; Gao et al., 2025; Li et al., 2024b) with a vast amount of text data collection, which poses challenges in generalizing such methods to MLLMs. Recently, COINCIDE (Lee et al., 2024) introduces a tiny 2B trained on the 665K target data pool to extract data embedding for the coreset (Sener & Savarese, 2018) selection. However, the use of the entire target dataset diminishes the significance of data selection. And high-dimensional embedding-based clustering and greedy sampling also pose scalability challenges. Despite the rarity of exploration for MLLMs, VLM research has proposed several multi-modal quality metrics that considers alignment as the objective (Maini et al., 2024; Gadre et al., 2023; Goyal et al., 2024). However, these scores do not necessarily correlate with optimal MLLM performance and may inadvertently select repetitive or redundant data points. Thus, balancing in-domain performance an cross-domain generalization still poses a great challenge for data selection studies. Built upon the pretrained target model, our obtained mmSSR can effectively follow the instruction of scoring and styling while the transferability of our fine-grained capabilities is well retained.

**LLM-Based Scoring**    employs a teacher model, such as proprietary ChatGPT (Brown et al., 2020; Achiam et al., 2023), as a cost-effective alternative to human annotation for scoring or ranking candidate instances. QuRating (Wettig et al., 2024) formulates four qualities regarding the quality of pretraining corpora, yet these qualities are investigated in isolation rather than being considered as composable. Deita (Liu et al., 2024b) defines the valuation of instructional data in terms of *quality* and *complexity*, and prompts ChatGPT (Achiam et al., 2023) to generate data that evolve in the two dimensions for training scorers. $DS^2$ directly prompts for scores in *rarity*, *complexity*, and *informativeness* for all candidate data points. We find that those high-level quality dimensions identified for LLM data are insufficient to capture the variability and inherent in multi-modal data concerning complex VL benchmarks. Furthermore, sample-level or pairwise scoring fails to account for global diversity, which is particularly crucial for SFT. And their chosen similarity-based thresholding are challenging to scale up. Our mmSSR is also built upon the judgments of a super model. However, we emphasize that the decomposition of model capabilities to the concrete level enriches the multi-modal data scoring while the style identification significantly simplify the selection procedure, enabling it to be customizable, effective, transferable, and scalable to the SOTA multi-modal open dataset.

# 3 METHOD

## 3.1 PROBLEM FORMULATION AND OVERVIEW

In this paper, we study the problem of data selection for MLLMs towards instruction alignment. Given a large-scale data pool $D = \{X_1, X_2, \ldots, X_N\}$, where each instance $x_i$ consists of multiple modalities, our task is to find a subset $D_{sel}$ of size $b$ that optimize the instruction following ability of the target model. Here we consider image and question-answer pairs, i.e., $X_i = (I_i, Q_i, A_i)$. The data budget $b$ is constrained by the computational budget of the SFT stage.

Our pipeline is built upon four key resources: a pretrained MLLM model ($\mathcal{M}_\theta$), a curated list of VL capabilities ($C$), a list of interaction styles ($S$) that support the instruction tuning of MLLMs, and a budget to assess a small amount of randomly sampled data $X'$. Our strategy includes four steps: (i) We employ a super model (e.g., proprietary GPT-4o or open-sourced Qwen-VL series) to generate judgments on $X'$, which encompasses the range of visual concepts in $C$ and assigns observed styles from the label space $S$; (ii) We finetune $\mathcal{M}_\theta$ with the subset and their corresponding scores and styles, yielding a series of $\mathcal{S}cr_i$ and $\mathcal{S}ty_i$; (iii) We infer on the whole data pool with respect to the rich capabilities and styles and perform style-aware top-score selection, yielding the selected subset $\hat{X}$ where $|\hat{X}| = b$; (iv) the pretrained model is efficiently finetuned with the subset. Once the mmSSR is obtained, within the domain of $D$, the composition of $\mathcal{S}cr_i$ can be customized towards general instruction tuning purposes or adapted for specialized requirements; one can also directly transfer mmSSR to new domain for data selection.

Next, we discuss the major contributions of our pipeline: formulating data quality valuation into *rich* and *transferable* capability criteria via scorers to build up MLLMs (Sec. 3.2), promoting data diversity via an instruction styler for efficient and *scalable* SFT (Sec. 3.3), and implementing style-aware, score-prioritized data selection (Sec. 3.4).

## 3.2 MULTI-MODAL RICH SCORERS

In the context of data valuation, especially for the instructional data, integrating advanced proprietary model, e.g., ChatGPT (Brown et al., 2020; Achiam et al., 2023), as a teacher has proven to be an effective automatic scoring approach given its high alignment with human preferences regarding conversation quality (Liu et al., 2024b; Pang et al., 2024; Wettig et al., 2024; Wang et al., 2024c; Yuan et al., 2024). A crucial aspect of this approach lies in the formulation of the scoring task, namely, formulating clear metrics and guidelines to instruct the model to query scores that are aligned with the optimization of MLLMs. We expect each instance-level score to exhibit clarity in multi-modal criteria, reliability in value and consistency across the entire data pool. However, we find that high-level, abstract keywords, such as quality, complexity (Pang et al., 2024; Liu et al., 2024b), accuracy and difficulty (Xu et al., 2023) adopted by previous selection methods for LLMs, fall short in capturing the complexity of our data with a greater variety of data modalities, a larger volume of data tokens, and a more heterogeneous pool of sources.

To overcome these challenges, we first enhance *clarity* by redefining the granularity of the scoring task, decomposing it into 14 specific capabilities, such as object spatial understanding and stem knowledge. These capabilities are both human-interpretable and model-attributable, covering rich visual-textual information (See Appendix A for the full list, examples and the decomposition process). Next, we query the scores for these criteria from the super model with corresponding brief explanations, to simplify the scoring task and instruct it to *align* with human understanding. To improve the *reliability* of the score value, we further request the super model to explain the rationale behind why a score is not higher or lower, in order to improve its answer in a self-reflection manner. As for cross-instance score *consistency*, to avoid overly lengthy prompt for VL inputs and changes in the finetuning paradigm, instead of prompting pairwisely, we specifically clarify the level of helpfulness for each value scale, which improves applicability across all capabilities and instances. Our prompt for scoring can be found in Appendix A.2.

To ensure cost-effectiveness, scalability and wide applicability, we randomly sample a small portion of data $X'$ (15% in our main experiment) from the target data pool to query the super model $\mathcal{G}$ for scores across all capabilities: $\mathcal{S}cr(X_i) = \{\mathcal{G}(I_i, Q_i, A_i; c_j)\}$, where $c_j$ is the $j$-th capability. The paired data-score instances $(X_i, \mathcal{S}cr(X_i))$ are then used to instruct a multi-modal model to predict

rich scores for all capability criteria. Thus, we can optimize the selection of comprehensive and general-purpose multi-modal datasets or construct specialized data mixtures as needed according to the judgments of our multi-modal rich scorers. In addition to the advantages of being rich, scalable, and customizable, our fine-grained decomposition of the scoring task ensures its transferability. As the adopted capabilities exhibit clear and general semantics, the scoring SFT task leverages the pre-trained model's understanding of these semantic capabilities through an interactive scoring process, rather than merely fitting to a limited amount of training data. When the capabilities are self-evident and consistent, the obtained fine-grained scorers exhibit strong generalizability.

### 3.3 Multi-modal Rich Styler

Data selection research for LLMs has revealed that data diversity is crucial, particularly during the supervised finetuning (SFT) stage (Zhou et al., 2024; Diddee & Ippolito, 2024; Xia et al., 2024b; Liu et al., 2024b; Pang et al., 2024; Li et al., 2024c). This challenge is further exacerbated by the heterogeneity of multi-modal data. For instance, the single-image training stage of LLaVA-OV (Li et al., 2024a) draws images from more than ninety different sources. To ensure selection diversity, existing studies derive the $D - dim$ *deep* feature as data representations, upon which the similarity computations and k-means greedy sampling are conducted within a complexity of $\mathcal{O}(NkD)$ (Liu et al., 2024b; Lee et al., 2024). Despite being straight-forward, the computationally burdensome strategy struggles to handle data of multi-modality in the magnitude of millions.

In light of the SAH (Zhou et al., 2024) that the main focus of SFT is to learn the interaction styles with users rather than acquiring new knowledge, we argue that the *superficial* styles can be a cheap and efficient proxy to capture interaction diversity. We curate a list of 9 styles observed in the current data pool (detailed in Tab. 3). Similar to the data curation for scorer training, we query the super model on the presence of each style $s_j$: $\mathcal{S}ty(X_i) = \{\mathcal{G}(I_i, Q_i, A_i; s_j)\}$, where $s_j$ is the $j$-th style. Then the data-style pairs $(X_i, \mathcal{S}ty(X_i))$ are used to instruct a model so as to infer rich styles on the entire data pool.

Compared to a large quantity of heuristic cluster centers ($k > 10,000$), utilizing concise and semantically rich data proxy (9 for mmSSR) enables us to efficiently bucket the data in $O(N)$ inference time, thereby avoiding the quadratic similarity calculations based on embeddings and the k-center hyperparameter tuning. The shift in perspective from traditional distribution-based sampling to style-based clustering not only ensures scalability as data continues to grow, but also directly facilitates the training objectives during the instruction tuning phase. Conversely, the effectiveness of the styler also demonstrates the applicability of SAH within the MLLM paradigm.

### 3.4 mmSSR for Data Selection

Given any set of capabilities of interest $\hat{C}$, the corresponding mmSSR are readily prepared to assess the candidate data pool. For each instance $X_i$ in $D$, mmSSR infers a score vector $\hat{\mathcal{S}}cr_i = \{r_{ic}\}$ where the score $r_{ic} \in [0, 1, \ldots, 5]$, and a style vector $\hat{\mathcal{S}}ty_i = \{g_{is}\}$ where the style membership is given by $g_{is} \in [0, 1]$. To achieve capability balancing and style diversity, we traverse the dataset in a Round-Robin fashion. Specifically, we define $|\hat{C}| \times |S|$ groups, and group $G_{cs} = \{i | r_{ic} > 0, g_{is} = 1\}$ is the set of indices of data points that belong to the group $cs$. Given a budget $b$, we iterate over each group for the highest-scored $\left\lfloor \frac{b}{|\hat{C}| \times |S|} \right\rfloor$ samples without replacement until the budget runs out:

$$D_{sel} = \bigcup^{|\hat{C}| \times |S|} d_{cs} \quad \text{where } |d_{cs}| = \left\lfloor \frac{b}{|\hat{C}| \times |S|} \right\rfloor + \delta_{cs}, \tag{1}$$

and $\delta_{cs}$ accounts for the remainder to ensure $\sum^{|\hat{C}| \times |S|} |d_{cs}| = b$.

To summarize, our mmSSR facilitates style-aware, score-prioritized sampling for multi-modal instructional data with efficiency and data scalability. Their formulation also guarantees transferability, customization and scalability in capabilities. We verify these features in the next experiment section.

# 4 EXPERIMENTS

## 4.1 EXPERIMENTAL SETUP

**Data pool.** We base our main experiments on the single-image SFT stage of LLaVA-OV (LLaVA-OVSI) (Li et al., 2024a), the current open-source, open-data SOTA MLLM series. Within its 3.2 million high-quality instances[1], 2.6 million multi-modal data are openly available, which we consider as the full dataset and perform sample selection on it. This well-curated dataset covers over 90 sources, encompassing natural images, math and reasoning questions, documents, charts, screenshots, and general OCR.

In our transfer experiments (Sec. 4.3), we use the earlier ShareGPT4V (Chen et al., 2024a) as the source data pool, which contains 624K image-question-answer pairs[2].

**Training setup.** For simplicity, we take the stage-1.5-7B checkpoint[3] provided by LLaVA-OneVision as the pretrained model for both mmSSR finetuning and single-image task model instruction tuning. To reduce the cost of comparative experiments, we decrease the maximum token length to 12k, ensuring that all training can be completed on 64 Nvidia H100 GPUs with a batch size of 128. Apart from this, all experimental settings strictly follow the training setup adopted by the official LLaVA-OneVision implementation.

In our transfer experiments (Sec. 4.3), we use the architecture of LLaVA-1.5-7B (Liu et al., 2024a) as the base model to instruct mmSSR. Likewise, the finetuning procedure of scorers and styler strictly follows the original implementation, all conducted on 8 Nvidia A100.

**Our setting.** Unless otherwise specified, we consider all capabilities, except for OCR, in our sampling experiments. We withhold the OCR capability to demonstrate the scalability of mmSSR on different capabilities, as presented in Sec. 4.4. In our experiments, we additionally make use of the 91 sources of LLaVA-OVSI data as subdomains and subdivide the grouped data, ensuring diversity among high-value samples across both language and visual modalities.

**Baselines.** We compare mmSSR with 8 methods across 6 different categories: a) *random sampling*: the strong diversity-prioritized baseline, evaluated based on the average results from three trials of different random splits; b) *perplexity*, including its two variants before (PPL-mid) and after (PPL-si) the single-image SFT on the entire data pool; c) *Deita* (Liu et al., 2024b)[4], the score and embedding-based SOTA methed for LLMs; d) *CLIP similarity* (Radford et al., 2021) (ViT-L) that evaluates the image-text alignment; e) *E5-V similarity* (Jiang et al., 2024), the SOTA MLLM-based universal embedding model built on LLaMA-3-8B (Dubey et al., 2024) that supports encoding longer textual sequences; and f) COINCIDE (Lee et al., 2024) and ICONS (Wu et al., 2024), the SOTA clustering-based selection strategy for MLLMs. To demonstrate the necessity of training proxy models, we directly prompt Qwen2-VL-7B (Wang et al., 2024b) and the fine-tuned LLaVA-OVSI checkpoint for scores and styles with the same instruction as used for GPT-4o.

**Evaluation benchmarks.** Under the VLMEvalKit (Duan et al., 2024) framework, we comprehensively evaluate our method on 14 multi-modal benchmarks, including MME (Fu et al., 2024a), MMBench$_{en-v1.1}$ (Liu et al., 2023a), MMStar (Chen et al., 2024b), MMMU (Yue et al., 2024), MMVet (Yu et al., 2023), BLINK (Fu et al., 2024b), MMT-Bench (Ying et al., 2024), AI2D (Kembhavi et al., 2016), ScienceQA (Lu et al., 2022), MathVista$_{MINI}$ (Lu et al., 2023). For the experiment in Sec. 4.4 that scales up in the OCR capability, we additionally evaluate mmSSR on OCRBench (Liu et al., 2023b), ChartQA (Masry et al., 2022), DocVQA (Mathew et al., 2021) and InfoVQA (Mathew et al., 2022). Since our setup focuses on the single-image SFT phase, the model does not possess the multi-image understanding ability. Thus, for MMMU and BLINK, we report results on the single-image QA split.

---

[1]https://huggingface.co/datasets/lmms-lab/LLaVA-OneVision-Data

[2]https://huggingface.co/datasets/Lin-Chen/ShareGPT4V

[3]https://huggingface.co/lmms-lab/llava-onevision-qwen2-7b-mid-stage-a4

[4]As Deita controls sample diversity through embedding similarity, the $O(N^2)$ complexity and the cost associated with threshold tuning is prohibitively expensive for scaling to our target data pool of 2.6 million instances. Therefore, we employ a variant that performs top-k sampling with its quality and complexity scores.

Table 1: Performance comparison on multi-modal benchmarks across varying budgets of 5, 10 and 30 of LLaVA-OVSI. We highlight the best result in **boldface** and underline the result if it beats the random baseline. The column >Rand reports the number of benchmarks where the method outperforms random sampling, and /FULL compares the performance of the selected data to that of the FULL dataset.

| | MMBench$_{en-v1.1}$ | MMStar | MMMU | MMVet | BLINK | MMT-Bench | MME | AI2D | ScienceQA | MathVista$_{mini}$ | >**Rand** | /**FULL** |
|---|---|---|---|---|---|---|---|---|---|---|---|---|
| **Budget: 5%** | | | | | | | | | | | | |
| Random | 73.74 | 47.98 | **43.70** | 42.34 | 50.61 | 58.87 | **2004.50** | 73.07 | 81.52 | 45.47 | - | 89.29 |
| PPL-mid | 67.34 | 45.27 | 38.98 | 30.18 | 45.27 | 54.33 | 1887.71 | 66.74 | 74.76 | 31.40 | 0/10 | 78.31 |
| PPL-si | 71.98 | 44.67 | 38.48 | 35.14 | 54.10 | 57.98 | 1856.79 | 67.84 | 78.24 | 36.50 | 1/10 | 83.10 |
| Deita | 72.91 | 47.47 | 41.28 | 40.23 | 52.59 | 56.57 | 1956.50 | 70.76 | 79.57 | 36.10 | 1/10 | 85.79 |
| CLIP | 74.23 | 47.27 | 40.08 | 35.73 | 52.96 | 56.73 | 1902.65 | 73.61 | 78.63 | 39.80 | 3/10 | 85.41 |
| E5-V | 70.90 | 43.00 | 38.78 | 38.44 | 49.94 | 54.65 | 1810.47 | 66.58 | 77.54 | 37.40 | 0/10 | 81.87 |
| COINCIDE | 72.76 | 48.33 | 43.17 | 45.60 | 49.43 | 57.53 | 1852.66 | 73.15 | 79.62 | 45.40 | 3/10 | 88.45 |
| ICONS | 66.72 | 52.20 | 41.18 | 38.03 | 47.92 | 55.96 | 1811.13 | 76.20 | 83.64 | 46.90 | 4/10 | 86.64 |
| mmSSR | **77.79** | **53.33** | 43.27 | 43.53 | 51.83 | **59.16** | 1938.68 | **77.66** | **88.45** | **52.00** | 8/10 | **93.20** |
| **Budget: 10%** | | | | | | | | | | | | |
| Random | 74.57 | 51.57 | 44.72 | 42.91 | 52.59 | 58.99 | 2033.28 | 74.42 | 84.33 | 47.80 | - | 91.70 |
| PPL-mid | 63.54 | 46.87 | 39.08 | 36.93 | 45.90 | 54.30 | 1831.03 | 67.23 | 73.87 | 39.50 | 0/10 | 80.72 |
| PPL-si | 74.69 | 49.80 | 41.28 | 40.60 | 53.09 | 57.95 | 1841.11 | 75.16 | 80.71 | 40.40 | 3/10 | 87.63 |
| Deita | 75.39 | 48.80 | 43.77 | 42.25 | 54.48 | 57.40 | 1996.34 | 71.60 | 78.33 | 40.80 | 2/10 | 88.72 |
| CLIP | 75.23 | 49.87 | 40.38 | 37.16 | 53.59 | 59.35 | 1921.04 | 76.62 | 80.07 | 41.00 | 4/10 | 87.69 |
| E5-V | 70.51 | 45.13 | 38.78 | 39.59 | 50.57 | 55.10 | 1787.94 | 68.94 | 77.54 | 37.20 | 0/10 | 82.76 |
| COINCIDE | 75.23 | 49.73 | 44.77 | 42.52 | 50.69 | 58.71 | 2027.58 | 74.77 | 82.05 | 47.00 | 3/10 | 90.66 |
| ICONS | 71.67 | 53.33 | 44.17 | 40.46 | 49.18 | 57.40 | 1789.60 | 76.65 | 85.23 | 51.10 | 4/10 | 89.91 |
| mmSSR | **77.32** | 53.27 | **45.06** | 42.98 | 54.10 | **59.61** | **2045.00** | **78.76** | **89.94** | **52.40** | 10/10 | **94.75** |
| **Budget: 30%** | | | | | | | | | | | | |
| Random | 78.25 | 54.60 | 44.40 | 46.10 | 55.23 | 59.61 | 2092.60 | 78.28 | 88.32 | 52.57 | - | 95.82 |
| PPL-mid | 73.99 | 54.93 | 43.97 | 41.01 | 53.09 | 58.78 | 2036.54 | 77.20 | 87.01 | 56.40 | 2/10 | 93.77 |
| PPL-si | 72.52 | 48.33 | 42.57 | 43.62 | 51.83 | 55.07 | 1976.46 | 76.55 | 78.48 | 42.20 | 0/10 | 88.22 |
| Deita | 76.93 | 54.13 | 43.67 | 44.04 | 55.11 | 59.66 | 2042.63 | 79.50 | 83.54 | 50.30 | 2/10 | 93.99 |
| CLIP | 74.30 | 53.80 | 43.07 | 45.87 | 51.95 | 59.16 | 2039.14 | 80.02 | 83.99 | 48.80 | 1/10 | 93.07 |
| E5-V | 74.30 | 46.07 | 43.27 | 47.80 | 50.32 | 57.85 | 1955.13 | 74.45 | 81.61 | 43.70 | 1/10 | 89.52 |
| COINCIDE | 78.02 | 55.47 | 45.66 | 46.24 | 52.84 | 59.80 | 2047.37 | 79.73 | 84.33 | 55.10 | 6/10 | 95.82 |
| ICONS | 71.90 | 53.40 | 43.87 | 42.25 | 50.32 | 59.23 | 1985.64 | 78.21 | 86.76 | 54.10 | 1/10 | 92.55 |
| mmSSR | **79.57** | **57.53** | 44.87 | **48.49** | **56.24** | **59.83** | **2132.93** | **81.25** | **92.46** | **57.40** | 10/10 | **99.11** |
| **FULL** | | | | | | | | | | | | |
| LLaVA-OVSI | 80.57 | 59.40 | 45.16 | 47.16 | 56.87 | 60.73 | 2117.56 | 81.87 | 92.76 | 59.60 | - | 100 |

## 4.2 MAIN RESULTS

**mmSSR consistently outperform competitors across varying data budget and benchmarks.** The comparative results on 10 multimodal benchmarks are presented in Tab. 1. It can be observed that whether the system is in a cold start (5% budget) or a warm start (30% budget) scenario, and regardless of the focus of the benchmark's evaluation, the samples identified by our mmSSR consistently outperform random sampling in most cases, making it an excellent choice in real-world applications. In contrast, other comparative methods fail to surpass random sampling under most of the benchmarks. Specifically, the mid-stage model of LLaVA-OV has not been instructed, hence the perplexity holds no referential significance. Alghough the SFT checkpoint LLaVA-OVSI shows marginally better performance, selecting samples with a fully fine-tuned model contradicts the motivation of the data selection task. Although the scorers of Deita (Liu et al., 2024b) have not been exposed to images, question-answer pairs should still aid in assessing sample value. However, results indicate that abstract criteria scoring like *quality* and *complexity* did not transfer well to the multi-modal task. While CLIP and E5-V can encode both modalities, experiments show that the emphasis of VLMs on image-text alignment is inconsistent with the optimization objectives of MLLMs. And COINCIDE (Lee et al., 2024) and COINS (Wu et al., 2024) shows vulnerability to larger and shifted data pool.

**Rich capabilities and styles guarantee the effectiveness of multi-modal data sampling.** In Tab. 2, we compare our mmSSR with rich capabilities and styles, noted as mmSSR(ich), to the mmSSP(oor) variant where we simply query GPT-4o's *quality* scores and corresponding explanations. In absence of style identification, we only improve diversity for mmSSP(oor) with image source during sampling. Results indicate that the abstract scoring criterion may introduce human-uninterpretable biases, which manifest as poor and inconsistent performance across different experimental settings and benchmarks.

**The superiority of mmSSR relies on its richness of criteria** rather than their exact composition.

Table 2: Ablation studies of mmSSR. mmSP(oor) is selected with primitive *quality* scores and data sources, mmScrR(ich) and mmStyR(ich) are based on rich scores and rich styles, respectively, while mmSSR leverages both. {method}${}_{\{source\}}$ indicates querying scores and styles from {source} to facilitate selection with {method}.

| | $R_{scr}$ | $R_{sty}$ | MMBench${}_{en-v1.1}$ | MMStar | MMMU | MMVet | BLINK | MMT-Bench | MME | AI2D | ScienceQA | MathVista${}_{mini}$ | >**Rand** | /**FULL** |
|---|---|---|---|---|---|---|---|---|---|---|---|---|---|---|
| **Budget: 5%** | | | | | | | | | | | | | | |
| Random | | | 73.74 | 47.98 | 43.70 | 42.34 | 50.61 | 58.87 | **2004.50** | 73.07 | 81.52 | 45.47 | - | 89.29 |
| mmSP${}_{GPT-4o}$ | | | 75.85 | 51.27 | 42.97 | **44.27** | 51.95 | 58.14 | 1940.27 | 73.61 | 81.46 | 45.00 | 5/10 | 90.14 |
| mmScrR${}_{GPT-4o}$ | ✓ | | 74.23 | 49.07 | 43.07 | 41.56 | 52.08 | 57.56 | 1996.27 | 74.77 | 80.12 | 44.80 | 4/10 | 89.17 |
| mmStyR${}_{GPT-4o}$ | | ✓ | 76.48 | 52.52 | 43.85 | 41.58 | 52.46 | 58.89 | 1945.68 | 76.64 | 86.88 | 49.87 | 8/10 | 92.07 |
| mmSSR${}_{Qwen2-VL}$ | ✓ | ✓ | 75.08 | 51.00 | 45.16 | 42.57 | 52.71 | 57.37 | 1955.78 | 74.74 | 84.88 | 48.90 | 8/10 | 91.37 |
| mmSSR${}_{LLaVA-OVSI}$ | ✓ | ✓ | 77.40 | 50.60 | 44.77 | 41.10 | 54.35 | 58.62 | 1952.97 | 75.81 | 87.75 | 40.40 | 6/10 | 90.68 |
| mmSSR${}_{GPT-4o}$ | ✓ | ✓ | **77.79** | 53.33 | 43.27 | 43.53 | 51.83 | **59.16** | 1938.68 | 77.66 | **88.45** | 52.00 | 8/10 | **93.20** |
| **Budget: 10%** | | | | | | | | | | | | | | |
| Random | | | 74.57 | 51.57 | 44.72 | 42.91 | 52.59 | 58.99 | 2033.28 | 74.42 | 84.33 | 47.80 | - | 91.70 |
| mmSP${}_{GPT-4o}$ | | | 77.24 | 50.40 | 44.27 | 42.52 | 53.47 | 59.48 | **2084.39** | 76.07 | 81.36 | 46.10 | 5/10 | 91.73 |
| mmScrR${}_{GPT-4o}$ | ✓ | | 73.76 | 49.40 | 44.77 | 42.80 | 46.91 | 57.24 | 2000.17 | 75.39 | 83.79 | 44.40 | 2/10 | 89.27 |
| mmStyR${}_{GPT-4o}$ | | ✓ | 77.72 | 54.36 | 44.17 | 44.62 | 54.05 | 59.60 | 1928.81 | 78.66 | 89.75 | **52.80** | 8/10 | 94.61 |
| mmSSR${}_{Qwen2-VL}$ | ✓ | ✓ | 76.24 | 53.33 | 44.87 | **45.60** | **55.11** | 59.16 | 2012.94 | 76.75 | 87.11 | 52.70 | 9/10 | 94.59 |
| mmSSR${}_{LLaVA-OVSI}$ | ✓ | ✓ | **77.79** | **54.40** | 44.67 | 42.02 | 54.98 | 58.23 | 2013.74 | **78.85** | 89.59 | 42.00 | 5/10 | 92.72 |
| mmSSR${}_{GPT-4o}$ | ✓ | ✓ | 77.32 | 53.27 | 45.06 | 42.98 | 54.10 | **59.61** | 2045.00 | 78.76 | 89.94 | 52.40 | 10/10 | **94.75** |
| **Budget: 30%** | | | | | | | | | | | | | | |
| Random | | | 78.25 | 54.60 | 44.40 | 46.10 | 55.23 | 59.61 | 2092.60 | 78.28 | 88.32 | 52.57 | - | 95.82 |
| mmSP${}_{GPT-4o}$ | | | 77.86 | 53.13 | **45.76** | 48.03 | 54.85 | 58.78 | 2050.69 | 78.92 | 86.91 | 55.80 | 4/10 | 96.31 |
| mmScrR${}_{GPT-4o}$ | ✓ | | 77.09 | 52.67 | 43.47 | 44.31 | 53.59 | 58.23 | 2024.57 | 79.11 | 87.90 | 52.20 | 1/10 | 93.93 |
| mmStyR${}_{GPT-4o}$ | | ✓ | 78.27 | 55.84 | 42.87 | 43.11 | 54.43 | 59.44 | 2079.25 | 80.42 | 92.15 | 55.96 | 5/10 | 96.07 |
| mmSSR${}_{Qwen2-VL}$ | ✓ | ✓ | 78.02 | 57.13 | 43.07 | 47.39 | 55.49 | **60.89** | 2096.60 | **81.64** | 90.28 | 57.40 | 8/10 | 97.91 |
| mmSSR${}_{LLaVA-OVSI}$ | ✓ | ✓ | 77.55 | 54.53 | 43.37 | 44.72 | 55.23 | 58.59 | 1980.48 | 81.02 | 91.87 | 49.60 | 2/10 | 94.73 |
| mmSSR${}_{GPT-4o}$ | ✓ | ✓ | **79.57** | **57.53** | 44.87 | 48.49 | **56.24** | 59.83 | **2132.93** | 81.25 | 92.46 | 57.40 | 10/10 | **99.11** |

We randomly sampled subsets of the criteria used in main experiments, reducing the set to 10 and even 6 criteria. The average results on the general benchmarks in Fig. 2 show that performance steadily drops when the number of criteria reduced, indicating that compromised richness degrades the informativeness and robustness of the selected subsets. Moreover, the stability with which a random subset of criteria consistently and significantly outperforms the random baseline indicates that the core of the proposed selection method lies in richness, rather than in a fixed combination of capabilities.

**Proxy mmSSR models trained using judgments from GPT-4o demonstrate the highest and most robust performance, while mmSSR selection with open-source MLLM judgments presents cost-effective alternatives.** Given the same prompt for rich capability scores and styles, as well as the same diversity-aware score-prioritized selection strategy, our mmSSR${}_{GPT-4o}$ fine-tunes proxy models to make predictions on the data pool, whereas mmSSR${}_{\{source\}}$ experiments leverage the instruction following ability of the source models to directly perform the scoring and styling task. As shown in Tab. 2, the specialized mmSSR models yield optimal performance, while utilizing the mmSSR selection method to directly query Qwen2-VL for rich scores and styles also promises a stable improvement compared to the best competitor. This further emphasizes the effectiveness of capability decomposition and style-based diversity sampling for the multi-modal data selection. And more importantly, we anticipate that mmSSR is well-positioned to benefit from future progress in both open-source and proprietary foundation models.

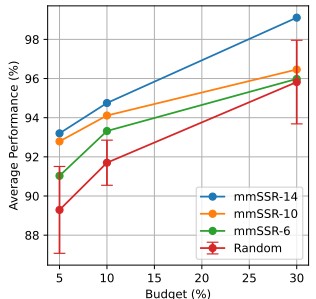

Figure 2: The critical role of richness: reduced richness degrades performance on randomly sampled subsets of 10 and 6 criteria.

## 4.3 TRANSFER IN DATA POOL AND SELECTION

**Transfer mmSSR from shareGPT4v to LLaVA-OVSI.** Common data curation scenarios often involve the addition of new subdomains. Here, we use 10% random subset of ShareGPT4V (Chen et al., 2024a) that consists of merely 12 sources (subdomains) as the base scenario to train mmSSR. These models are then directly generalize to LLaVA-OVSI (Li et al., 2024a) of 91 sources for inference and sampling. Results illustrated in Fig. 3 demonstrate strong generalization capability to the larger data pool with open sources and novel knowledge.

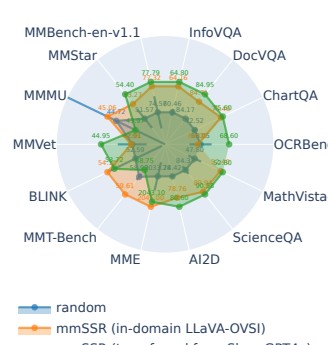

Figure 3: Transferability of mmSSR models: trained on Share-GPT4v data, directly inferences on large-scale LLaVA-OVSI pool.

**Transfer mmSSR selection to a different model.** We also expect the selected subset to be generally applicable, instead of being dependent on specific architecture or training settings (Munjal et al., 2020). To verify the effectiveness of the subset selected by mmSSR that are finetuned from LLaVA-OVSI-7B, we use it to train a 0.5B LLaVA-OVSI model. Results in Fig. 4 with 5% budget show that the superiority of our method remain, demonstrating strong robustness.

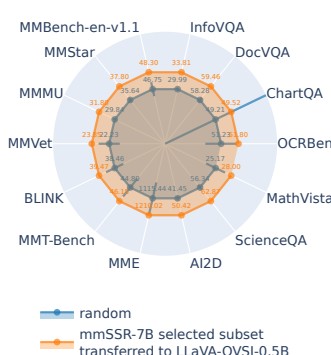

Figure 4: Transferability of mmSSR subsets: selected by mmSSR-7B, directly used to train a 0.5B variant.

The superior performance of mmSSR can be attributed to the generality of the rich capabilities and styles we have articulated, which is effective across different model architectures, datasets, and validation settings. Specifically, *scores and styles are more generalizable than task model responses*. While model-based methods rely on their specific model responses (e.g., perplexity, embeddings and influence) for data valuation, our mmSSR is instructed to score and identify instructional styles characterized by general and explainable semantics. Furthermore, *rich scores and styles are more generalizable than coarse-grained quality-like descriptors*. For pretrained MLLMs to be finetuned, while the understanding of quality might shift, the intrinsic knowledge of fine-grained capabilities and styles is more readily shared and transferable. Thus, the finetuned mmSSR and the selected subsets consistently guarantee strong and robust performance.

### 4.4 SCALABILITY IN DATA QUANTITY AND CAPABILITY

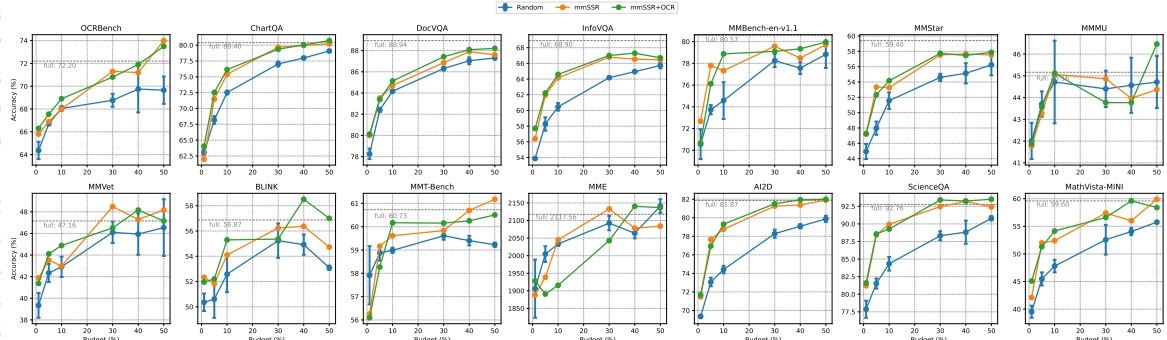

Figure 5: Results of scaling in data quantity (1% → 50%) and data capability (13 capabilities + OCR).

We further validated the scalability of the proposed mmSSR under varying data volumes in both colder-start (1%) and hot-start (40%, 50%) scenarios, achieving consistently superior MLLM performance, as shown in Fig. 5. Beyond quantity, we consider a data expansion scenario commonly encountered in real-world applications, scaling up the capability dimension within the existing data pool. Taking OCR for example, we query judgments on it and fine-tune mmSSR to select highly-scored OCR samples. The newly added samples lead to steady improvements in OCR-related benchmarks. Furthermore, they contribute to the growth of general benchmarks or sustain advantageous positions, demonstrating great scalability.

## 5 CONCLUSIONS

mmSSR leverages the nature of instruction tuning to decompose multi-modal data into capability scores and interaction styles and make judgment over those proxies. It facilitates diversity-aware score-prioritized sampling, demonstrating superior performance across 14 benchmarks and 6 budget settings. Furthermore, the formulation of concrete quality and style criteria with semantics guarantees capability customization, strong generalizability, and efficient scaling potential, which promises broad applicability and accessibility.

## 6 ETHICS STATEMENT

All authors have read and adhered to the ICLR Code of Ethics. We have upheld high standards of scientific excellence by presenting our methods transparently to encourage reproducibility. We have carefully considered the broader societal impacts of our research, striving to contribute positively to human well-being while taking proactive steps to avoid harm, unfairness, and discrimination. We have diligently credited the intellectual contributions of others. Our work is intended to promote the responsible and ethical advancement of machine learning.

## 7 REPRODUCIBILITY STATEMENT

We have included the complete source code in the supplementary materials. Meantime, we guarantee the self-contained nature of the paper with a detailed description of our experimental setup, covering the dataset, the model, training configurations, and evaluation benchmarks, provided in Sec. 4.1. Further implementation details, including the derivation of capabilities and styles, and the prompt template to obtain rich scores and styles, are available in the appendix (Sec. A and A.2).

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

## A  ADDITIONAL IMPLEMENTATION DETAILS

### A.1  PROMPT TEMPLATE TO DETERMINE CAPABILITIES AND STYLES

Tab. 3 presents capabilities and interaction styles that we identified from the current open-access instruction-following datasets for building general-purpose MLLMs. The identification process was iterative and heuristic, comprising three main stages. We first discovered an initial set of capabilities by analyzing the LLaVA-OVSI data sources. Next, to expand this set and ensure comprehensive coverage, we prompted GPT-4o with random samples to uncover novel patterns. Finally, we performed two iterations of refinement, merging semantically similar items and pruning long-tail capabilities. The prompt used for this process is provided below. An identical methodology was used to determine the interactive styles.

We emphasize that the efficacy of mmSSR is not rigidly tied to this specific composition. As demonstrated in Sec. 4.2, its success depends on the richness of the capability subspace. This suggests that as data grows—either through new public datasets or private specialized sources—our pipeline can be readily adapted to extract a broader range of valuable knowledge, such as for image-based creative writing, chain-of-thought reasoning, and solving competition-level mathematical problems.

Table 3: 14 criteria we recognize as the foundational pillars for developing vision perception and reasoning capabilities within MLLMs, and the interaction styles we identify from instructional multi-modal data.

| mm Capabilities | Definitions | Examples |
|---|---|---|
| activity recognition | actions or behaviors of humans, animals, or objects | Fig. 14 |
| causal reasoning | cause-and-effect relationships between events or variables to predict outcomes and explain phenomena | Fig. 15 |
| humanities | history, literature, philosophy, art, and culture to understand human experiences and societal developments | Fig. 16 |
| STEM knowledge | science, technology, engineering, and mathematics, chemistry, economics etc | Fig. 17 |
| comparative analysis | compare multiple entities, concepts, or datasets to identify similarities, differences, and relationships | Fig. 18 |
| data understanding | documents, tables, charts, graphics, infographics | Fig. 19 |
| object spatial understanding | the positions, orientations, countings and relationships of objects | Fig. 20 |
| attribute identification | various characteristics and properties of objects, such as identity, color, size, shape, material, emotion, and other distinguishing features | Fig. 21 |
| logical deduction | to analyze information, recognize patterns, draw valid conclusions based on structured principles of logic and make reasoned decisions | Fig. 22 |
| scene understanding | complex environment with objects, their attributes, spatial relationships, and activities, as well as surrounding information and circumstances within the scene | Fig. 23 |
| fine-grained recognition | subtle differences and specific features within similar categories of objects | Fig. 24 |
| language generation | generate coherent and contextually appropriate text in various languages, styles, and formats based on instructions | Fig. 25 |
| in-context learning | follow the demonstrations of the task within a given conversation | Fig. 26 |
| optical character recognition | the conversion between images of printed/handwritten text and machine-readable text | Fig. 27 |
| style | multi-choice, coordinate, yes/no, word/short-phrase, short description, detailed description, comparison, chain-of-thought (step-by-step), specified style | Fig. 14-27 |

## Prompt to Derive the Capabilities of Interest

```
You are an AI expert tasked with defining the essential capabilities for a
next-generation multi-modal large language model. To achieve this, we will follow a
structured, three-step process: Discovery, Expansion, and Refinement.

Stage 1: Discovery
First, I want you to act as a research analyst. Your task is to analyze the following
data sources to identify a broad set of foundational capabilities and iterative tasks
suggested by the data.

Data Sources: [
    COCO Caption,
    Vision FLAN,
    ...
]ᵃ

Based on your analysis of these sources, generate a list of potential capabilities.ᵇ

Step 2: Expansion
Now, I want you to think beyond this initial list. I will provide you with a list of
initial capability candidates and random samples of visual question answer pairs. Your
task is to think creatively and identify any new capabilities that are not adequately
covered by the existing list but are necessary for answering these visual questions.

Initial List:
{initial capabilities}

Random Samples:
{Random samples drawn from the datapool}

Your goal is to heuristically expand our list. Think creatively and do not worry
about overlap at this stage; focus on generating a comprehensive set of potential
new capabilities.ᶜ

Step 3: Refinement
Next, I want you to refine our expanded list of capabilities. I will provide you with a
list of capabilities. Your task is to identify and merge criteria that are semantically
similar or redundant. For each proposed merger, provide a brief justification. The goal
is to create a comprehensive yet more concise and semantically coherent list.

Expanded List:
{expanded capabilities}

Present the new, merged list.ᵈ

Next, I want you to analyze the frequency and importance of each capability on the
merged list. Your task is to analyze the merged list of capabilities below and identify
capabilities that are the most critical for a wide variety of multi-modal questions.
Also, identify the potentially long-tailed datapoints, which might be capabilities
that are too niche, rarely required, or could be considered a sub-component of other
capability we've already defined.

Merged List:
{merged capabilities}

Your goal is to create a final, core set of capabilities for training while discarding
less representative ones. Please provide the final list of essential capabilities and a
separate list you chose to set aside.ᵉ
```
_______________________________

[a] See Tab. 5 for the full list.

[b] The initial list generated by the LLM in this round will be denoted as {initial capabilities}.

[c] This step can be iterated multiple times to get a sufficient candidate list. The expanded list generated in this round will be denoted as {expanded capabilities}.

[d] The merged list suggested in this round will be denoted as {merged capabilities}.

[e] This step can be iterated multiple times to get a refined list.

## A.2 Prompt Template for Rich Scores and Styles

Below gives our query template, where {Input} and {Response} are paired questions and answers. Multi-round user-assistant interactions are concatenated for demonstration. To enhance the stability of pointwise scoring of by GPT-4o, we define a score scale from 0 to 5 and establish clear benchmarks. Each data sample is evaluated on all capability dimensions, querying scores and recalling all observed styles in the text modality. To improve the self-consistency of responses, we require explanations for the given scores. Particularly, in the valuation of multi-modal data, we emphasize the importance of balancing the correlation between image and text modalities in task-specific contexts, i.e., scoring and styling, rather than allowing the model to be biased towards a single modality, such as being dominated by language or vision.

To examine the effectiveness of our prompt and the quality of GPT-4o judgments, for each capability, we present examples scored 0-5 in Fig. 14-Fig. 27, accompanied by detailed explanations. We note that when obtaining costly human scoring is impractical, using MLLMs for annotations could introduce hallucinations (e.g., 5th example of Fig. 23, 4th example of Fig. 26). However, it still serves as a viable sub-gold standard. Take the 3rd and 5th samples in Fig. 27 for example: Although the visual content in these scenes is similar, the text queries focus on distinct elements. When the task requires generating an "informative summary" and the answer is related to reading text on a vehicle, the contribution of this training sample to the OCR capability is crucial, yielding a score of 3. Conversely, when the task shifts to global scene understanding with an emphasis on road details, the background text in the 5th image becomes irrelevant, resulting in a score of 0. Hallucinations present in the original samples within the answers, such as the 4th example in Fig. 20, can also be identified and thus given lower scores, preventing the propagation of incorrect information in subsequent SFT processes. These cases demonstrate the efficiency of prompt instructions, highlighting that the balance between image-query-task in data curation meets expectations.

---

**Prompt to Query GPT-4o for Rich Scores and Styles**

```
System Prompt:
You are an AI expert rater designed to analyze the Visual Question Answering (VQA)
instance in the user query to perform the following tasks step-by-step:
Step 1: Classify the VQA instance into given conversation style.
Step 2: Evaluate the helpfulness of the information provided in the VQA instance with
respect to various model capabilities. Specifically, rate how well this information
could enhance each capability of a multi-modal large language model through learning
from it.
Step 3: Output the results strictly follow the JSON format.

User Prompt:
## Instruction
You need to perform the following three steps to rate the User Query and output result
in the dictionary format.
Step 1: Classify the instance in interaction style. Determine the task style of the VQA
instance and select styles from the list ``task_styles" below. Sort the selected styles
by frequency of occurrence.
Step 2: Rate each capability from 0-5. For each capability listed and explained in
``task_capabilities" below, analyze how effectively the VQA instance could enhance
that capability of a Multimodal Large Language Model (MLLM) by learning from it. Rate
each capability using the scores from the ``score_scale" list below in refernce to the
guidelines. Please ensure that the scores are well-distributed across the range.
Finally, output the results strictly following the dictionary format defined in Output
Format. Do not output any additional tokens outside it.

## User Query
Question: {Input}
Answer: {Response}

## Task Styling
task_styles = [
    multi-choice,
    coordinate,
    ...
]ᵃ
```

---
ᵃSee Tab. 3 for the full list.

```
Prompt to Query GPT-4o for Rich Scores and Styles

## Task Capabilities
task_capabilities = [
    optical character recognition,   # the conversion between images of
printed/handwritten text and machine-readable text
    ...
]ᵃ

## Rating Scale
score_scale = [
    0,   # Not Relevant: The VQA instance does not present or relate to the capability
in any meaningful way.
    1,   # Minimal: The VQA instance offers very little information relevant to the
capability, providing negligible value for enhancement.
    2,   # Fair: The VQA instance contains some relevant information but lacks depth and
clarity, contributing minimally to the model's learning in this capability.
    3,   # Good: The VQA instance provides a fair amount of relevant information, which
can moderately aid in the model's learning and enhancement of the capability.
    4,   # Significant: The VQA instance offers substantial information that is highly
relevant and beneficial, significantly aiding the model's learning and enhancement of
the capability.
    5,   # Excellent: The VQA instance is exceptionally rich in relevant information,
providing comprehensive and clear insights that would greatly enhance the model's
learning and mastery of the capability.
]

## Output Format

    {
        "style": "<list of string>",
        "capability2score": "<dict of str:int>",
        "capability2explanation": "<dict of str:str>",
    }

    ____________________________
    ᵃSee Tab. 3 for the full list and explanation.
```

## B  VALIDATION ON BENCHMARK TEST SPLITS

We compared the baselines and our mmSSR across various selection settings within a unified framework, obtaining batches of results. The intra-batch comparison of results is sufficient to validate the effectiveness of sampling strategies. Thus, considering the limitations on the number of evaluations in the online assessment system, we primarily report the results of MMBench$_{en-v1.1}$ (Liu et al., 2023a), MMMU (Yue et al., 2024), and MMT-Bench (Ying et al., 2024) on their validation set in Tab. 1. We then select the top samplers for submission to the online test split, with the results presented in Table 4.

Similar to the main experiment, our comparative results on the test split consistently outperform or match the performance of SOTA baselines. In addition to maintaining a stable absolute advantage regardless of the data budget, mmSSR exhibits particularly remarkable effectiveness during the challenging cold phase. Without dependency on pre-trained model features or pre-selected hyperparameters, our semantic-based rich capabilities and superficial styles show stronger transferability to the test sets. Besides, the trends and the full performance observed in the MMMU dataset indicate that the task remains challenging for the current single-image data pool. To achieve further improvements, it would be beneficial to integrate additional external data that includes college-level multidisciplinary knowledge.

## C  EFFICIENCY COMPARISON

Fig. 6 shows the wall-clock time cost analysis of multi-modal data selection methods and corresponding average performance on LLaVA-VOSI compared to full fine-tuning. For each trial, the time consumption is measured from the start of the selection model training to the completion of the task model training with budgeted data. It can be observed that mmSSR significantly outperforms the baselines in both accuracy and speed. While full fine-tuning requires 62.35 hours on 64 H100 GPUs, our method achieves 93.20%, 94.75%, and 99.11% of the final performance within 30.4, 31.8 and 36.8 hours if staring from the scratch, when training from pipeline scratch.

Table 4: Performance comparison on the benchmarks with online test splits conducted across varying budgets of 5%, 10% and 30% of LLaVA-OVSI. We highlight the best result in **boldface** and underline the result if it beats the random baseline. The column >Rand presents the number of benchmarks where the method exceeds random sampling, and /FULL compares the performance of sampled data with that of the FULL dataset.

| | MMBench$_{en-v1.1-test}$ | MMMU$_{test}$ | MMT-Bench$_{all}$ | >Rand | /FULL |
|---|---|---|---|---|---|
| **Budget: 5%** | | | | | |
| Random | 73.45 | 40.37 | 59.98 | - | 96.32% |
| Deita | 73.97 | 36.30 | 56.57 | 1/3 | 91.40% |
| COINCIDE | 73.09 | 40.30 | 57.47 | 0/3 | 94.74% |
| ICONS | 69.04 | 37.30 | 57.49 | 0/3 | 90.63% |
| mmSSR | **75.84** | **41.30** | **60.10** | 3/3 | **98.15**% |
| **Budget: 10%** | | | | | |
| Random | 74.55 | 40.40 | 60.54 | - | 97.12% |
| Deita | 75.17 | 37.00 | 57.40 | 1/3 | 92.92% |
| COINCIDE | 74.44 | 40.40 | 58.68 | 1/3 | 96.05% |
| ICONS | 71.90 | 38.80 | 58.66 | 1/3 | 93.68% |
| mmSSR | **76.05** | **40.90** | **60.68** | 3/3 | **98.23**% |
| **Budget: 30%** | | | | | |
| Random | 77.33 | **41.13** | 59.59 | - | 98.36% |
| Deita | 76.88 | 40.00 | 59.56 | 0/3 | 97.24% |
| COINCIDE | **78.18** | 41.00 | 59.77 | 2/3 | 98.71% |
| ICONS | 72.21 | 40.40 | 59.70 | 1/3 | 95.67% |
| mmSSR | 78.13 | 41.10 | **59.80** | 2/3 | **98.78**% |
| **FULL** | | | | | |
| LLaVA-OVSI | 79.27 | 41.40 | 60.70 | - | 100% |

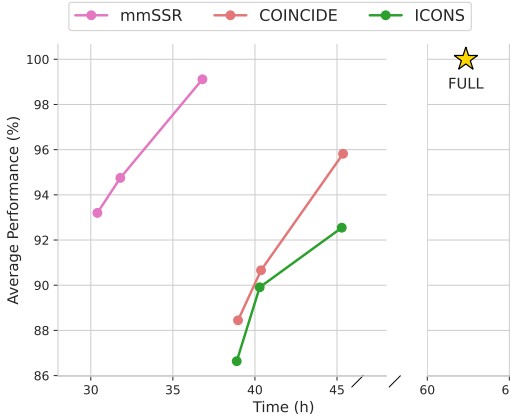

Figure 6: Comparison of selection methods on average performance and time cost for MLLM finetuning.

Note that in our pipeline, the model training and scoring are performed only once, and adapting to different budget settings requires less than one minute for ranking. In contrast, COINCIDE (Lee et al., 2024) requires over one hour for clustering and selection on million-scale datasets each time. And ICONS (Wu et al., 2024) necessitates gradient computation on the validation data for each evaluation benchmark, as well as similarity estimation to the candidate data, making its computational cost difficult to estimate.

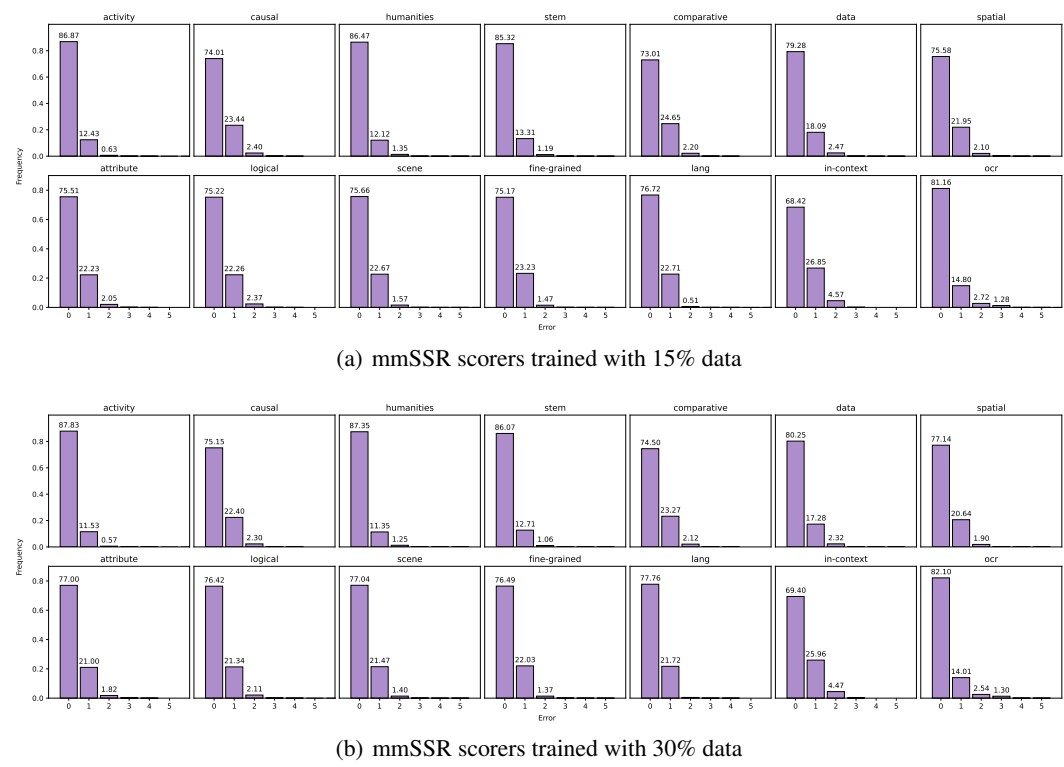

(a) mmSSR scorers trained with 15% data

(b) mmSSR scorers trained with 30% data

Figure 7: The mean absolute error of mmSSR scorer predictions against GPT-4o judgment over 14 capabilities.

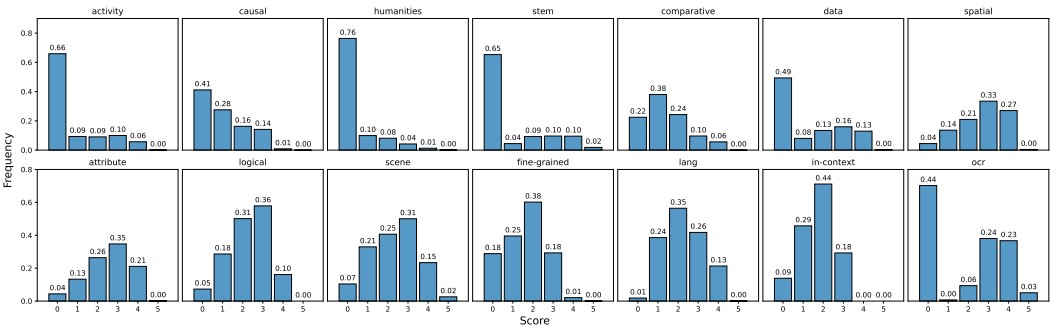

Figure 8: Distribution of scores of 14 capabilities across the LLaVA-OVSI dataset inferred by mmSSR.

# D VALIDATION OF SCORER AND STYLER PREDICTIONS

## D.1 ERROR ANALYSIS OF SCORER

The training of Scorers and Styler uses 15% of the LLaVA-OVSI data, following the original instructional tuning strategy (Li et al., 2024a)[5]. To minimize the exploration cost of mmSSR in practical applications, no hyperparameter fine-tuning is introduced in the pipeline. In this section, to verify the performance of the mmSSR judgments, we additionally annotated the remaining 85% of the single-image data pool with GPT-4o as a validation set. The mean absolute error (MAE) of the scorer is shown in Fig. 7(a). Overall, across 14 capabilities with varying levels of granularity and differentiation difficulty, an average of 77.7% of the scores are exactly the same as those given by

---

[5]https://github.com/LLaVA-VL/LLaVA-NeXT/blob/main/scripts/train/finetune_si.sh

GPT-4o. When allowing a margin of error of 1 in scoring, the accuracy reached 97.8%, which is a reasonable relaxation, considering that GPT's pointwise judgment is not a definitive gold standard and may inherently contain fluctuations (Wettig et al., 2024).

Based on the score distribution shown in Fig. 8, we observe that the accuracy of identifying rare and specialized abilities, such as those in the humanities and STEM fields, is relatively high, particularly in recognizing their absence. Consequently, in diversity-oriented sampling, such minority data are seldom overlooked. In contrast, while more ubiquitous abilities exhibit a normal or uniform distribution, giving completely identical scores is more challenging. In fact, if we randomly verify samples with closely related yet different scores, we observe that the differences in their values are often indistinguishable to human evaluators. For instance, in Fig. 22, the difference between scores of 1 and 2 in logical reasoning for the 4th example is minimal. Similarly, in Fig. 18, the distinction between values of 5 and 4 in comparative analysis for the 1st example is also minor.

We further increase the GPT-4o annotated data volume to 30% of the total dataset to train scorer. MAE results in Fig. 7(b) demonstrate a marginal performance improvement compared to models trained with 15% data, validating that the scoring models we derive has undergone sufficient training.

Thus, in summary, our mmSSR demonstrates the capability to deliver reliable and justified assessments when confronted with unseen multi-modal data.

## D.2 ERROR ANALYSIS OF STYLER

In Fig. 9, we present the precision and recall distributions of the styler against the GPT-4o recognition. Compared to the scoring task, determining the interaction style present in conversations is straightforward and yields higher accuracy. The average precision across the whole data pool reached 96.35%, while the recall achieved 95.80%.

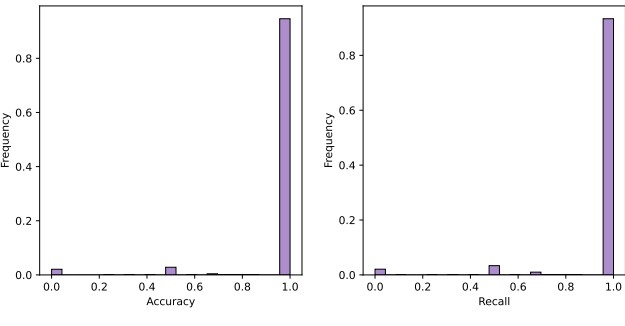

(a) Precisions of mmSSR styler    (b) Recalls of mmSSR styler

Figure 9: The precision and recall of mmSSR styler predictions against GPT-4o judgment among 9 styles.

## D.3 VISUALIZATION OF CAPABILITY SCORES AND STYLER

For each capability of interest, we group the data based on GPT-4o's scoring range of 0-5, randomly sample within each score group. Image-text pairs, GPT-4o scores, style recognition and explanations, and our mmSSR judgments are shown in Fig. 14-Fig. 27. The correspondence between capabilities and visualizations is detailed in Tab. 3.

## E ANALYSIS OF SELECTED DATA

## E.1 SCORES OF SELECTED DATA

To illustrate the information obtained by our sampler, in Fig. 10, we present the score distributions of the selected subsets, focusing on two different sampling ratios: 10% and 30%. The scores used

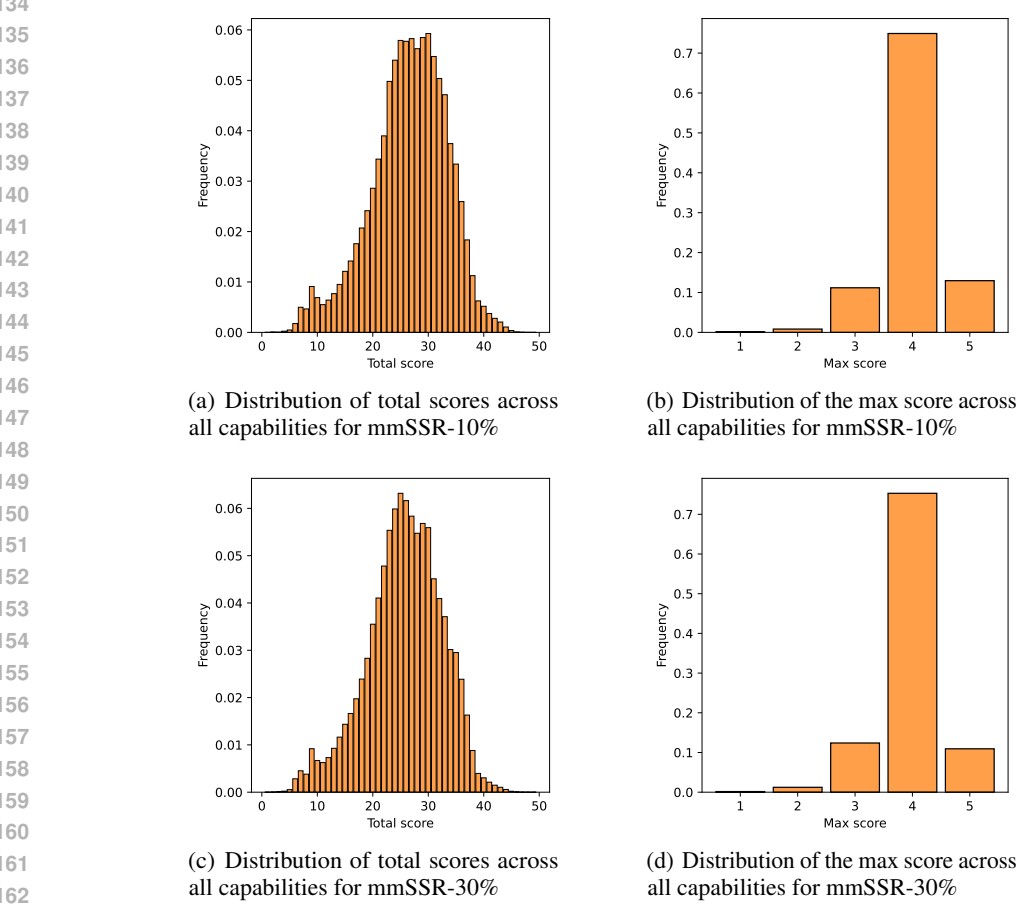

(a) Distribution of total scores across all capabilities for mmSSR-10%

(b) Distribution of the max score across all capabilities for mmSSR-10%

(c) Distribution of total scores across all capabilities for mmSSR-30%

(d) Distribution of the max score across all capabilities for mmSSR-30%

Figure 10: Score distribution analysis of the selected mmSSR-10% and mmSSR-30%.

in the statistics are derived from the evaluations of our mmSSR trained with 15% scoring data. The distribution of total scores across all capabilities, as depicted in Fig. 10(a) and Fig. 10(c), manifests a bell-shaped curve. This characteristic shape is predominantly attributed to the limited availability of high-scoring options, which inherently restricts the sampler's ability to select from the upper echelon of scores. Consequently, the distribution gravitates towards the central scores, forming a normal distribution pattern.

A notable aspect of our sampling approach is the selection of low scores despite their relatively modest total score contributions. Since selection is executed through a round-robin sampling methodology, which prioritizes minority yet specialized capabilities that have low synergy with other capabilities, such as STEM, which is critical for addressing niche challenges of benchmarks. The inclusion of these capabilities enhances the diversity and robustness of the sampled subset, ensuring that our model is equipped to handle a broad spectrum of scenarios.

Fig. 10(b) and Fig. 10(d) further corroborate the sampler's behavior, illustrating the distribution of maximum scores across all capabilities. The concentration of scores around the mid-range (specifically, scores of 4) underscores the mmSSR's tendency to opt for samples with higher information efficiency.

By incorporating both highly-valued and specialized mid-range capabilities, our mmSSR not only ensures a balanced representation of capabilities but also reinforces the sampler's capacity to enhance the overall performance and adaptability of the model.

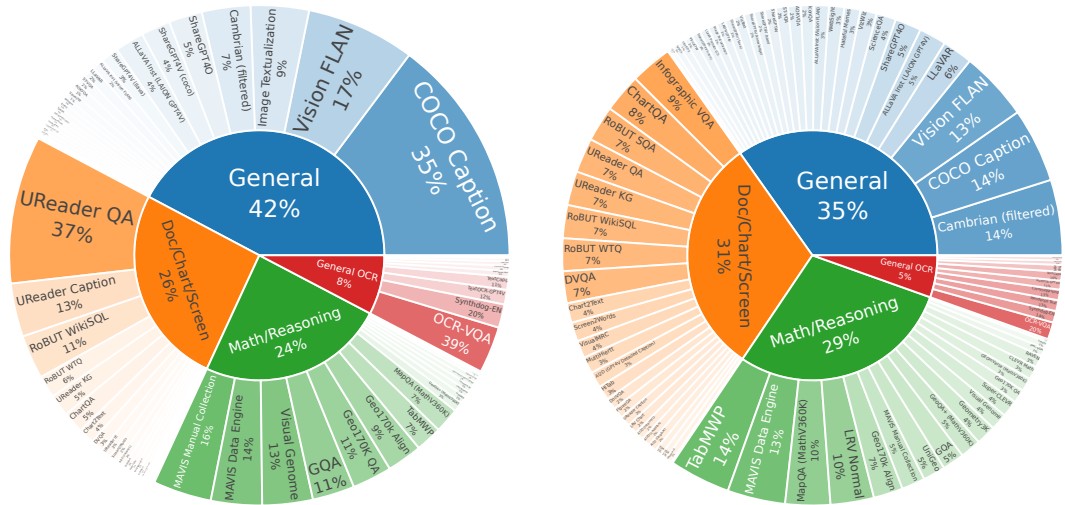

Figure 11: Data source statistics of the original LLaVA-OVSI data pool (L) and our mmSSR-10% (R).

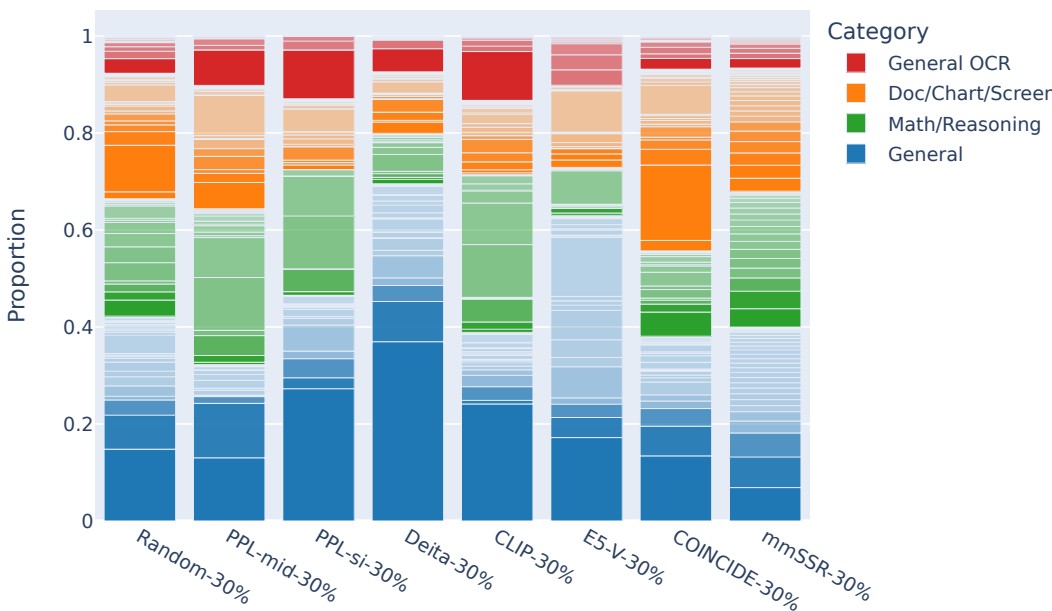

Figure 12: Comparison of data source statistics between our selection and those of competitors. For brevity, the figure displays only a small subset of the legends. Data sources of the same category are represented by shared color schemes, in accordance with Fig. 11.

## E.2 SOURCES OF SELECTED DATA

Considering the heterogeneity of multi-modal data sources and the challenges posed by extensive and comprehensive evaluation benchmarks, it is crucial to promote diversity in the instruction finetuning stage. Following the original data hierarchy Li et al. (2024a), we detail the statistical information of the full data pool and our sampled 10% data in Tab. 5, and illustrate it in Fig. 11.

The subset reveal a shift towards balance when employing the proposed mmSSR. The original LLaVA-OVSI on the left, is dominated by the General category, which constitutes 42% of the data, in which COCO Caption makes up 35%. In contrast, the subset on the right, sampled with mmSSR,

shows a more balanced source distribution. Here, the General category is reduced to 35%, while COCO Caption decreases to 14%. Notably, the Math/Reasoning category expands from 24% to 29% in the sampled subset, and the Doc/Chart/Screen category increases from 26% to 31%. Fig. 12 highlights the differences between comparative methods and ours. Notably, mmSSR exhibits a more balanced distribution across various sources, while Deita and E5-V embedding shows a pronounced concentration in the dominant general data, PPL and CLIP favor math/reasoning data, especially Visual Genome, over others, and COINCIDE is skewed towards Doc/Chart/Screen. The effective reallocation of training data underscore the advantages of mmSSR in achieving a more equitable representation of data sources, enhancing the robustness of the fine-tuned model in general instruction-following tasks and improving its adaptability for more challenging tasks, such as mathematical problem-solving and infographic reasoning.

### E.3 STYLES OF SELECTED DATA

Likewise, we provide a comparative analysis of the data style distributions in Fig. 13. As can be seen, our mmSSR exhibits a distinct distribution pattern characterized by a balanced representation of several key styles. This distribution indicates a comprehensive and balanced coverage of styles that are essential for the SFT stage, thereby enhancing the robustness of the finetuned model. In comparison, other sampling methods show a skewed distribution, with certain styles, like *detailed description* that usually contributes more training tokens, and *yes/no* or *word/short-phrase* that is ubiquitous in benchmarks, being overrepresented. The imbalance could potentially limit the versatility and applicability of the datasets generated by these methods. Notably, our approach achieves a more equitable distribution across different styles, including *comparison* and *chain-of-thought*, which are crucial for reasoning tasks. This balanced distribution is indicative of our method's capability to cater to a broader range of machine learning applications, thereby positioning our sampling method as a versatile tool for dataset curation.

In summary, the analysis in Fig. 10, 11, 12 and 13 demonstrates that mmSSR can provide a highly informative subset over rich capabilities, which enjoying a well-rounded and diverse dataset composition over both data sources and instruction styles, contributing to the data efficiency and explainability of MLLMs.

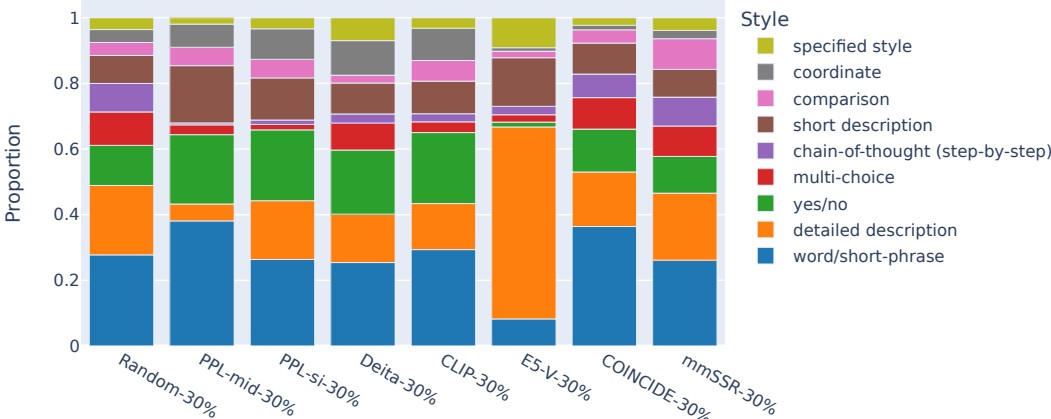

Figure 13: Comparison of data style statistics between our selection and those of competitors.

Table 5: Number of samples and proportions of sources for LLaVA-OVSI and mmSSR selected subsets.

| Source | LLaVA-OVSI | | mmSSR-10% | | mmSSR-30% | |
|---|---|---|---|---|---|---|
| | # Samples | Prop. | # Samples | Prop. | # Samples | Prop. |
| General | | | | | | |
| COCO Caption | 391219 | 34.97% | 12828 | 13.83% | 54707 | 17.21% |
| Vision FLAN | 186060 | 16.63% | 11922 | 12.85% | 50068 | 15.75% |
| Image Textualization | 99573 | 8.90% | 1523 | 1.64% | 7760 | 2.44% |
| Cambrian (filtered) | 83125 | 7.43% | 12997 | 14.01% | 39517 | 12.43% |
| ShareGPT4O | 57284 | 5.12% | 4533 | 4.89% | 15400 | 4.84% |
| ShareGPT4V (coco) | 50017 | 4.47% | 1510 | 1.63% | 9755 | 3.07% |
| ALLaVA Inst (LAION GPT4V) | 49990 | 4.47% | 4696 | 5.06% | 9976 | 3.14% |
| ShareGPT4V (llava) | 29990 | 2.68% | 1613 | 1.74% | 9768 | 3.07% |
| ALLaVA Inst (Vision FLAN) | 19990 | 1.79% | 2577 | 2.78% | 9220 | 2.90% |
| LLaVAR | 19790 | 1.77% | 5231 | 5.64% | 19509 | 6.14% |
| ST-VQA | 17242 | 1.54% | 2096 | 2.26% | 7013 | 2.21% |
| AOKVQA | 16534 | 1.48% | 2188 | 2.36% | 7486 | 2.35% |
| Visual7W | 14361 | 1.28% | 1478 | 1.59% | 4802 | 1.51% |
| WebSight | 9995 | 0.89% | 2632 | 2.84% | 8742 | 2.75% |
| VisText | 9964 | 0.89% | 1769 | 1.91% | 6363 | 2.00% |
| TallyQA | 9868 | 0.88% | 1126 | 1.21% | 7309 | 2.30% |
| ShareGPT4V (sam) | 8990 | 0.80% | 1862 | 2.01% | 8451 | 2.66% |
| Hateful Memes | 8495 | 0.76% | 2765 | 2.98% | 8495 | 2.67% |
| LAION GPT4V | 8048 | 0.72% | 1525 | 1.64% | 7139 | 2.25% |
| LLaVA Pretrain LCS | 6989 | 0.62% | 1512 | 1.63% | 6580 | 2.07% |
| VizWiz | 6604 | 0.59% | 2809 | 3.03% | 5220 | 1.64% |
| ScienceQA | 5932 | 0.53% | 3388 | 3.65% | 5930 | 1.87% |
| IconQA | 2496 | 0.22% | 2214 | 2.39% | 2496 | 0.79% |
| ShareGPT4V (knowledge) | 1988 | 0.18% | 1770 | 1.91% | 1988 | 0.63% |
| ShareGPT4V | 1926 | 0.17% | 1911 | 2.06% | 1926 | 0.61% |
| InterGPS | 1275 | 0.11% | 1275 | 1.37% | 1275 | 0.40% |
| CLEVR | 700 | 0.06% | 700 | 0.75% | 700 | 0.22% |
| VQARAD | 308 | 0.03% | 308 | 0.33% | 308 | 0.10% |
| Doc/Chart/Screen | | | | | | |
| UReader QA | 252954 | 36.96% | 5962 | 7.27% | 21233 | 10.51% |
| UReader Caption | 91434 | 13.36% | 1784 | 2.18% | 6861 | 3.40% |
| RoBUT WikiSQL | 74984 | 10.95% | 5688 | 6.94% | 20290 | 10.05% |
| RoBUT WTQ | 38241 | 5.59% | 5622 | 6.86% | 15094 | 7.47% |
| UReader KG | 37550 | 5.49% | 5872 | 7.16% | 21335 | 10.56% |
| ChartQA | 36577 | 5.34% | 6669 | 8.14% | 18550 | 9.18% |
| Chart2Text | 26956 | 3.94% | 3403 | 4.15% | 8751 | 4.33% |
| DVQA | 22000 | 3.21% | 5489 | 6.70% | 17219 | 8.52% |
| UReader IE | 17322 | 2.53% | 1060 | 1.29% | 3307 | 1.64% |
| Screen2Words | 15725 | 2.30% | 3256 | 3.97% | 9225 | 4.57% |
| AI2D (InternVL) | 12403 | 1.81% | 1530 | 1.87% | 6715 | 3.32% |
| DocVQA | 10194 | 1.49% | 1999 | 2.44% | 5272 | 2.61% |
| RoBUT SQA | 8509 | 1.24% | 6148 | 7.50% | 8509 | 4.21% |
| Infographic VQA | 8489 | 1.24% | 7233 | 8.82% | 8489 | 4.20% |
| MultiHiertt | 7614 | 1.11% | 2855 | 3.48% | 7614 | 3.77% |
| AI2D (GPT4V Detailed Caption) | 4864 | 0.71% | 2746 | 3.35% | 4864 | 2.41% |
| AI2D (Original) | 3247 | 0.47% | 1457 | 1.78% | 3247 | 1.61% |
| VisualMRC | 3022 | 0.44% | 3021 | 3.69% | 3022 | 1.50% |
| HiTab | 2495 | 0.36% | 2495 | 3.04% | 2495 | 1.24% |
| AI2D (cauldron) | 2429 | 0.35% | 1499 | 1.83% | 2429 | 1.20% |
| VSR | 2152 | 0.31% | 1062 | 1.30% | 2152 | 1.07% |
| FigureQA | 1880 | 0.27% | 1880 | 2.29% | 1880 | 0.93% |
| LRV Chart | 1776 | 0.26% | 1776 | 2.17% | 1776 | 0.88% |
| TQA | 1366 | 0.20% | 1177 | 1.44% | 1366 | 0.68% |
| Diagram Image2Text | 295 | 0.04% | 295 | 0.36% | 295 | 0.15% |

| Source | LLaVA-OVSI | | mmSSR-10% | | mmSSR-30% | |
|---|---|---|---|---|---|---|
| | # Samples | Prop. | # Samples | Prop. | # Samples | Prop. |
| Math/Reasoning | | | | | | |
| MAVIS Manual Collection | 99990 | 15.60% | 4033 | 5.22% | 15763 | 7.08% |
| MAVIS Data Engine | 87348 | 13.63% | 9964 | 12.90% | 30124 | 13.52% |
| Visual Genome | 86417 | 13.49% | 2895 | 3.75% | 14992 | 6.73% |
| GQA | 72140 | 11.26% | 3771 | 4.88% | 13056 | 5.86% |
| Geo170K QA | 67823 | 10.58% | 2330 | 3.02% | 10150 | 4.56% |
| Geo170k Align | 60242 | 9.40% | 5067 | 6.56% | 12398 | 5.56% |
| TabMWP | 45169 | 7.05% | 10516 | 13.61% | 28677 | 12.87% |
| MapQA (MathV360K) | 42637 | 6.65% | 7735 | 10.01% | 21827 | 9.80% |
| GeoQA+ (MathV360K) | 17162 | 2.68% | 3578 | 4.63% | 16106 | 7.23% |
| UniGeo | 11949 | 1.86% | 3855 | 4.99% | 11947 | 5.36% |
| LRV Normal | 10489 | 1.64% | 7481 | 9.68% | 10489 | 4.71% |
| Geometry3K | 9724 | 1.52% | 3415 | 4.42% | 9724 | 4.36% |
| GEOmVerse (MathV360K) | 9298 | 1.45% | 2326 | 3.01% | 9029 | 4.05% |
| Super-CLEVR | 8642 | 1.35% | 2883 | 3.73% | 6774 | 3.04% |
| CLEVR Math | 5280 | 0.82% | 2248 | 2.91% | 5248 | 2.36% |
| RAVEN | 2100 | 0.33% | 2100 | 2.72% | 2100 | 0.94% |
| Geo3k | 2091 | 0.33% | 1165 | 1.51% | 2091 | 0.94% |
| PMC-VQA | 1798 | 0.28% | 1387 | 1.80% | 1797 | 0.81% |
| GEOS | 498 | 0.08% | 498 | 0.64% | 498 | 0.22% |
| General OCR | | | | | | |
| OCR-VQA | 80000 | 39.29% | 2942 | 20.29% | 15556 | 29.56% |
| Synthdog-EN | 40093 | 19.69% | 2006 | 13.84% | 8730 | 16.59% |
| TextOCR-GPT4V | 25104 | 12.33% | 1528 | 10.54% | 8013 | 15.23% |
| TextCaps | 21942 | 10.78% | 1520 | 10.48% | 7262 | 13.80% |
| Rendered Text | 9995 | 4.91% | 1881 | 12.97% | 2796 | 5.31% |
| ChromeWriting | 8825 | 4.33% | 1813 | 12.50% | 3277 | 6.23% |
| HME100K | 7450 | 3.66% | 761 | 5.25% | 3364 | 6.39% |
| IAM | 5658 | 2.78% | 1167 | 8.05% | 2151 | 4.09% |
| K12 Printing | 2567 | 1.26% | 710 | 4.90% | 982 | 1.87% |
| IIIT5K | 1990 | 0.98% | 171 | 1.18% | 486 | 0.92% |

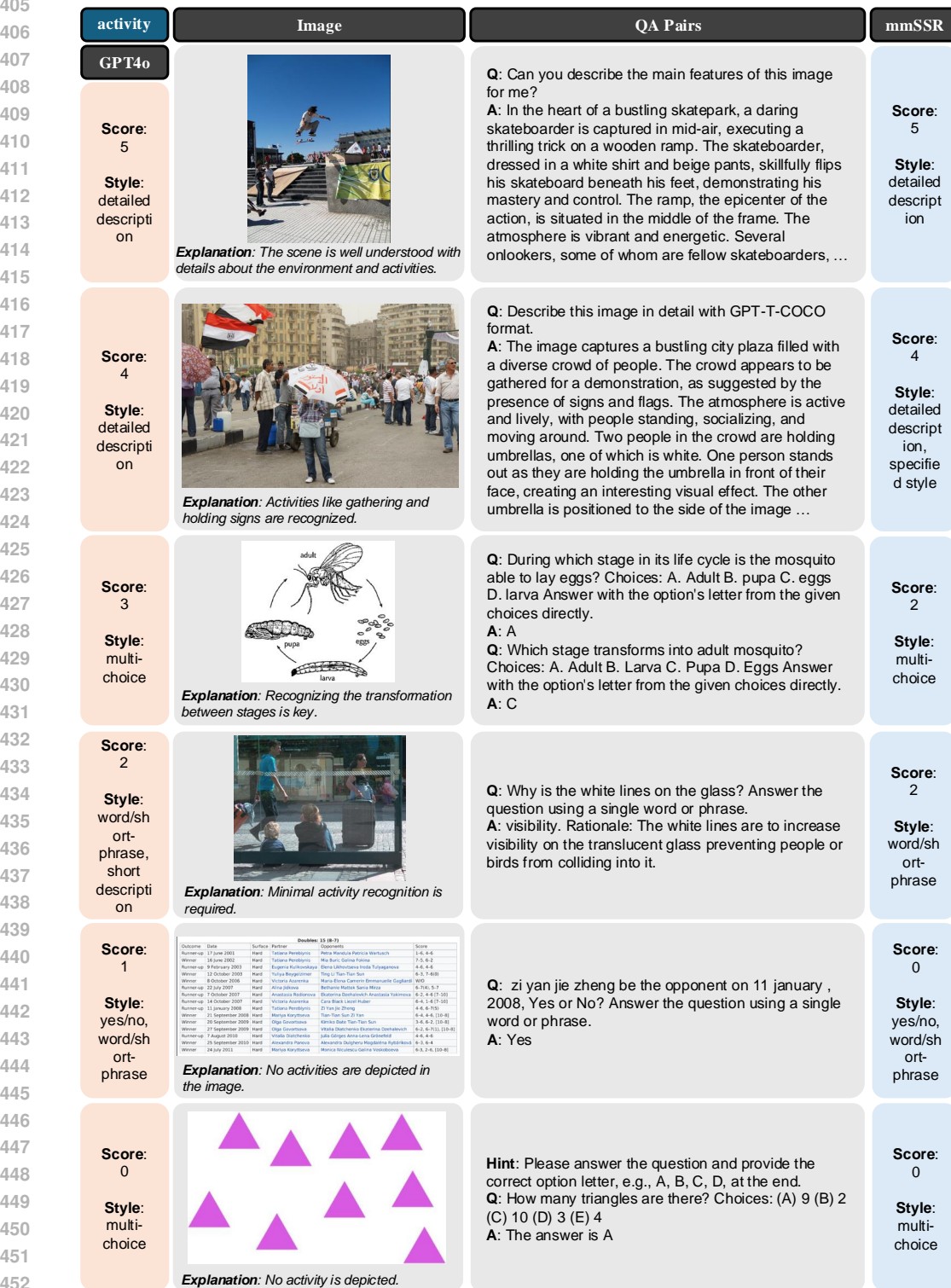

Figure 14: Randomly selected samples from the LLaVA-OVSI data pool within each score group for the **activity recognition** capability, ranging from 5 to 0, as evaluated by GPT-4o, along with corresponding score explanations and mmSSR predictions.

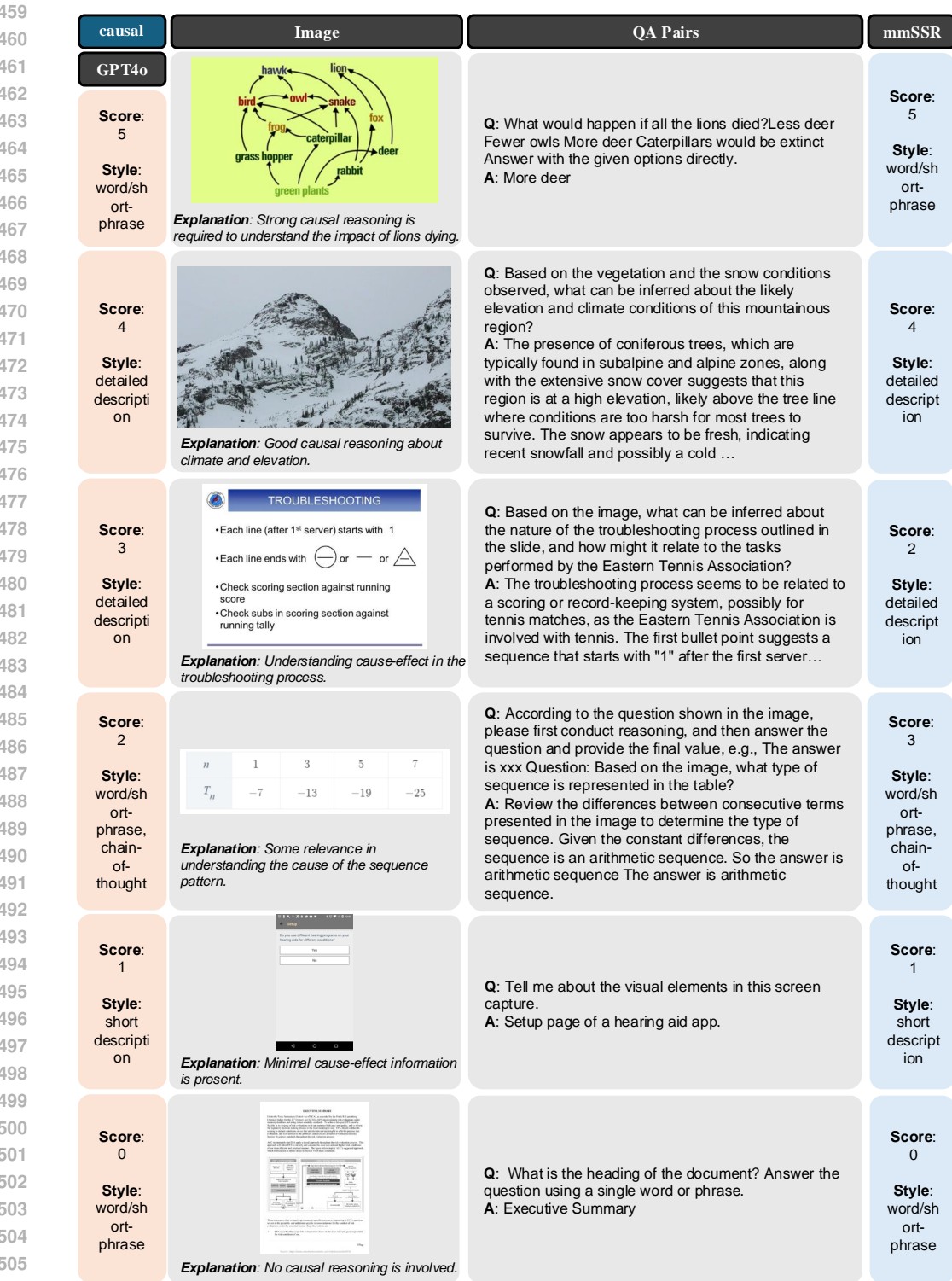

Figure 15: Randomly selected samples from the LLaVA-OVSI data pool within each score group for the **causal reasoning** capability, ranging from 5 to 0, as evaluated by GPT-4o, along with corresponding score explanations and mmSSR predictions.

| humanities | Image | QA Pairs | mmSSR |
|---|---|---|---|
| **GPT4o**

**Score**: 5

**Style**: detailed description | *Explanation: Strong understanding of mythology and cultural context is demonstrated.* | **Q**: What is the role of the solitary figure with the pole in the dynamics of this scene?
**A**: Based on the image and the title, the solitary figure with the pole is most likely Charon, the ferryman of Hades, who in Greek and Roman mythology, transports the souls of the newly deceased across the rivers Styx and Acheron that divided the world of the living from the world of the dead. His authoritative stance and detachment from the chaos suggest he is overseeing the embarkation of the souls to their final destination in the afterlife, indifferent to their pleas and suffering. | **Score**: 5

**Style**: detailed description |
| **Score**: 4

**Style**: short description | *Explanation: The historical nature of the engraving is relevant to humanities.* | **Q**: Can you describe the visual appearance of this image?
**A**: Early 16th-century engraving of Toulouse | **Score**: 4

**Style**: short description |
| **Score**: 3

**Style**: word/short-phrase, short description | *Explanation: Involves literature, contributing to humanities understanding.* | **Q**: What is the title of the book mentioned in the image?
**A**: The title of the book is "Llama Llama Red Pajama and 19 Other Favorites" by Anna Dewdney.
**Q**: Who are the narrators of the book?
**A**: The book is read by the author, Anna Dewdney, and Cassandra Campbell.
**Q**: Can you list some stories or titles mentioned in the image?
**A**: Some of the titles mentioned are: Llama Llama Red Pajama, Llama Llama Mad at Mama, Llama Llama Misses Mama, Llama Llama Holiday Drama, … | **Score**: 3

**Style**: word/short-phrase, short description |
| **Score**: 2

**Style**: detailed description | *Explanation: Some cultural context related to media events is present.* | **Q**: Considering the consistent background and the title indicating a cast interview, what can be inferred about the nature of the event where these photos were taken, and how do the individuals' attire and expressions contribute to the setting's informality or formality?
**A**: The nature of the event can be inferred to be a promotional interview for the television show or movie "SERVANT." The consistency in the background across all six photos suggests they were taken at the same venue, likely set up for media interviews… | **Score**: 2

**Style**: detailed description |
| **Score**: 1

**Style**: yes/no, word/short-phrase | *Explanation: Minimal cultural context is provided by the meme format.* | **Q**: Can this meme be considered disrespectful? Answer the question using a single word or phrase.
**A**: No. | **Score**: 1

**Style**: yes/no, word/short-phrase |
| **Score**: 0

**Style**: detailed description | *Explanation: No humanities-related content is present.* | **Q**: What's happening in the scene?
**A**: In the image, a man clad in a vibrant red and black motorcycle suit stands next to a matching red and black motorcycle, prominently displayed in the center of the image. The motorcycle, leaning on its side stand, is parked in front of a red garage door adorned with multiple windows. The man, holding a helmet in his left hand, appears to be inspecting the motorcycle with a keen eye, standing very close to the camera… | **Score**: 0

**Style**: detailed description |

Figure 16: Randomly selected samples from the LLaVA-OVSI data pool within each score group for the **humanities** capability, ranging from 5 to 0, as evaluated by GPT-4o, along with corresponding score explanations and mmSSR predictions.

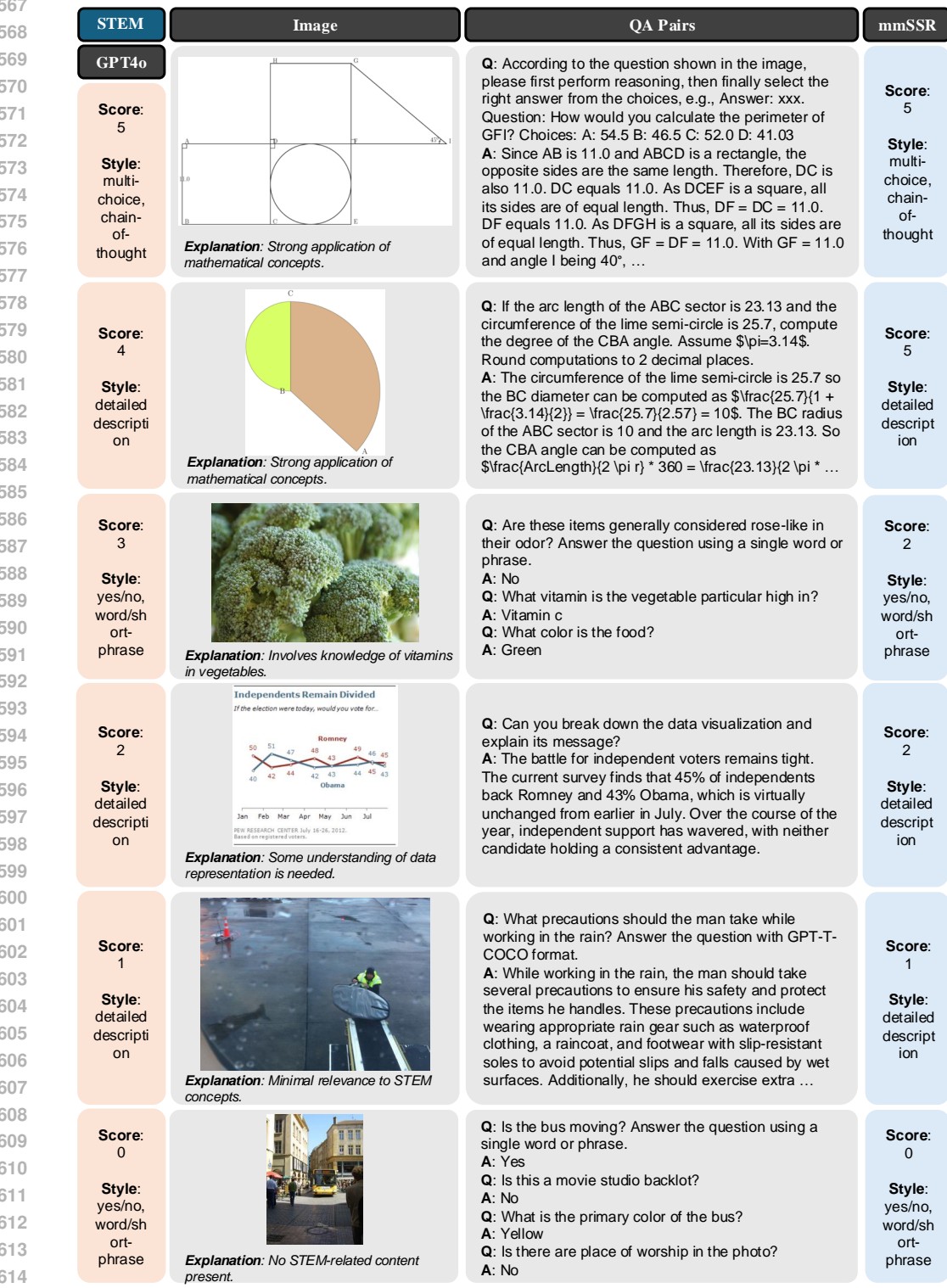

| STEM | Image | QA Pairs | mmSSR |
|---|---|---|---|
| **GPT4o** | | **Q**: According to the question shown in the image, please first perform reasoning, then finally select the right answer from the choices, e.g., Answer: xxx. Question: How would you calculate the perimeter of GFI? Choices: A: 54.5 B: 46.5 C: 52.0 D: 41.03 **A**: Since AB is 11.0 and ABCD is a rectangle, the opposite sides are the same length. Therefore, DC is also 11.0. DC equals 11.0. As DCEF is a square, all its sides are of equal length. Thus, DF = DC = 11.0. DF equals 11.0. As DFGH is a square, all its sides are of equal length. Thus, GF = DF = 11.0. With GF = 11.0 and angle I being 40°, … | **Score**: 5 **Style**: multi-choice, chain-of-thought |
| **Score**: 5 **Style**: multi-choice, chain-of-thought *Explanation: Strong application of mathematical concepts.* | | | |
| **Score**: 4 **Style**: detailed description | *Explanation: Strong application of mathematical concepts.* | **Q**: If the arc length of the ABC sector is 23.13 and the circumference of the lime semi-circle is 25.7, compute the degree of the CBA angle. Assume $\pi=3.14$. Round computations to 2 decimal places. **A**: The circumference of the lime semi-circle is 25.7 so the BC diameter can be computed as $\frac{25.7}{1 + \frac{3.14}{2}} = \frac{25.7}{2.57} = 10$. The BC radius of the ABC sector is 10 and the arc length is 23.13. So the CBA angle can be computed as $\frac{ArcLength}{2 \pi r} * 360 = \frac{23.13}{2 \pi * …$ | **Score**: 5 **Style**: detailed description |
| **Score**: 3 **Style**: yes/no, word/short-phrase | *Explanation: Involves knowledge of vitamins in vegetables.* | **Q**: Are these items generally considered rose-like in their odor? Answer the question using a single word or phrase. **A**: No **Q**: What vitamin is the vegetable particular high in? **A**: Vitamin c **Q**: What color is the food? **A**: Green | **Score**: 2 **Style**: yes/no, word/short-phrase |
| **Score**: 2 **Style**: detailed description | *Explanation: Some understanding of data representation is needed.* | **Q**: Can you break down the data visualization and explain its message? **A**: The battle for independent voters remains tight. The current survey finds that 45% of independents back Romney and 43% Obama, which is virtually unchanged from earlier in July. Over the course of the year, independent support has wavered, with neither candidate holding a consistent advantage. | **Score**: 2 **Style**: detailed description |
| **Score**: 1 **Style**: detailed description | *Explanation: Minimal relevance to STEM concepts.* | **Q**: What precautions should the man take while working in the rain? Answer the question with GPT-T-COCO format. **A**: While working in the rain, the man should take several precautions to ensure his safety and protect the items he handles. These precautions include wearing appropriate rain gear such as waterproof clothing, a raincoat, and footwear with slip-resistant soles to avoid potential slips and falls caused by wet surfaces. Additionally, he should exercise extra … | **Score**: 1 **Style**: detailed description |
| **Score**: 0 **Style**: yes/no, word/short-phrase | *Explanation: No STEM-related content present.* | **Q**: Is the bus moving? Answer the question using a single word or phrase. **A**: Yes **Q**: Is this a movie studio backlot? **A**: No **Q**: What is the primary color of the bus? **A**: Yellow **Q**: Is there are place of worship in the photo? **A**: No | **Score**: 0 **Style**: yes/no, word/short-phrase |

Figure 17: Randomly selected samples from the LLaVA-OVSI data pool within each score group for the **stem knowledge** capability, ranging from 5 to 0, as evaluated by GPT-4o, along with corresponding score explanations and mmSSR predictions.

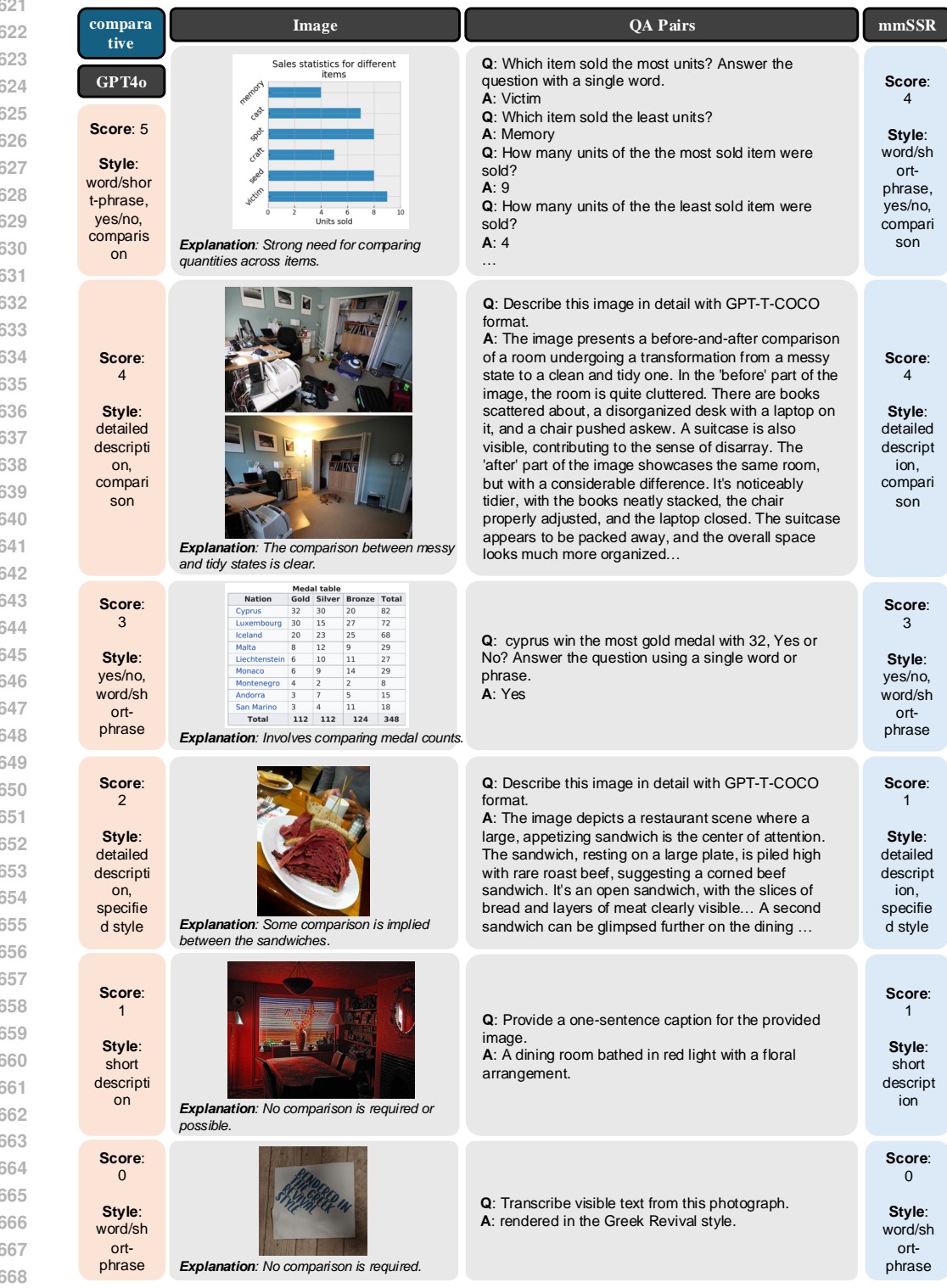

Figure 18: Randomly selected samples from the LLaVA-OVSI data pool within each score group for the **comparative analysis** capability, ranging from 5 to 0, as evaluated by GPT-4o, along with corresponding score explanations and mmSSR predictions.

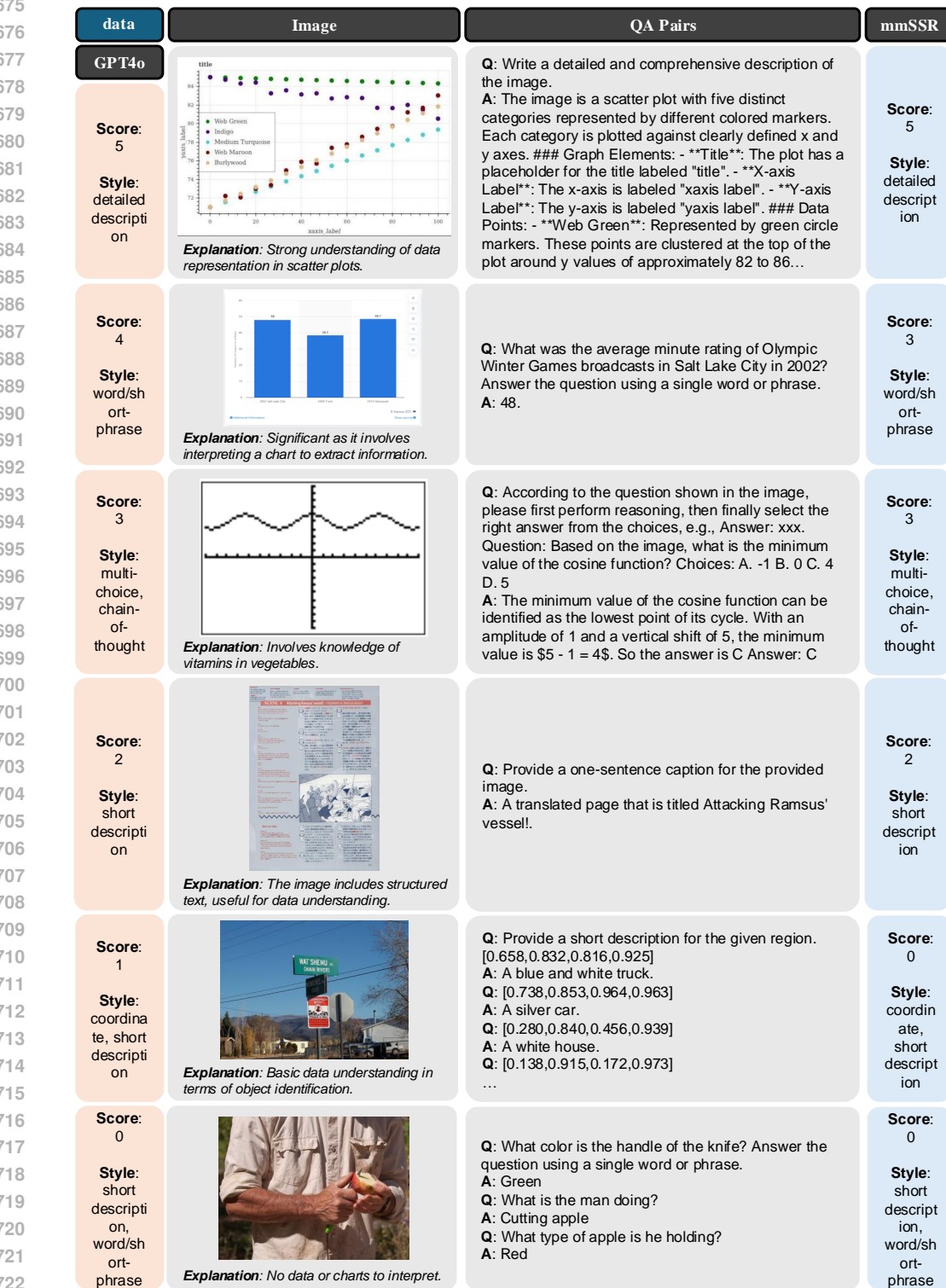

Figure 19: Randomly selected samples from the LLaVA-OVSI data pool within each score group for the **data understanding** capability, ranging from 5 to 0, as evaluated by GPT-4o, along with corresponding score explanations and mmSSR predictions.

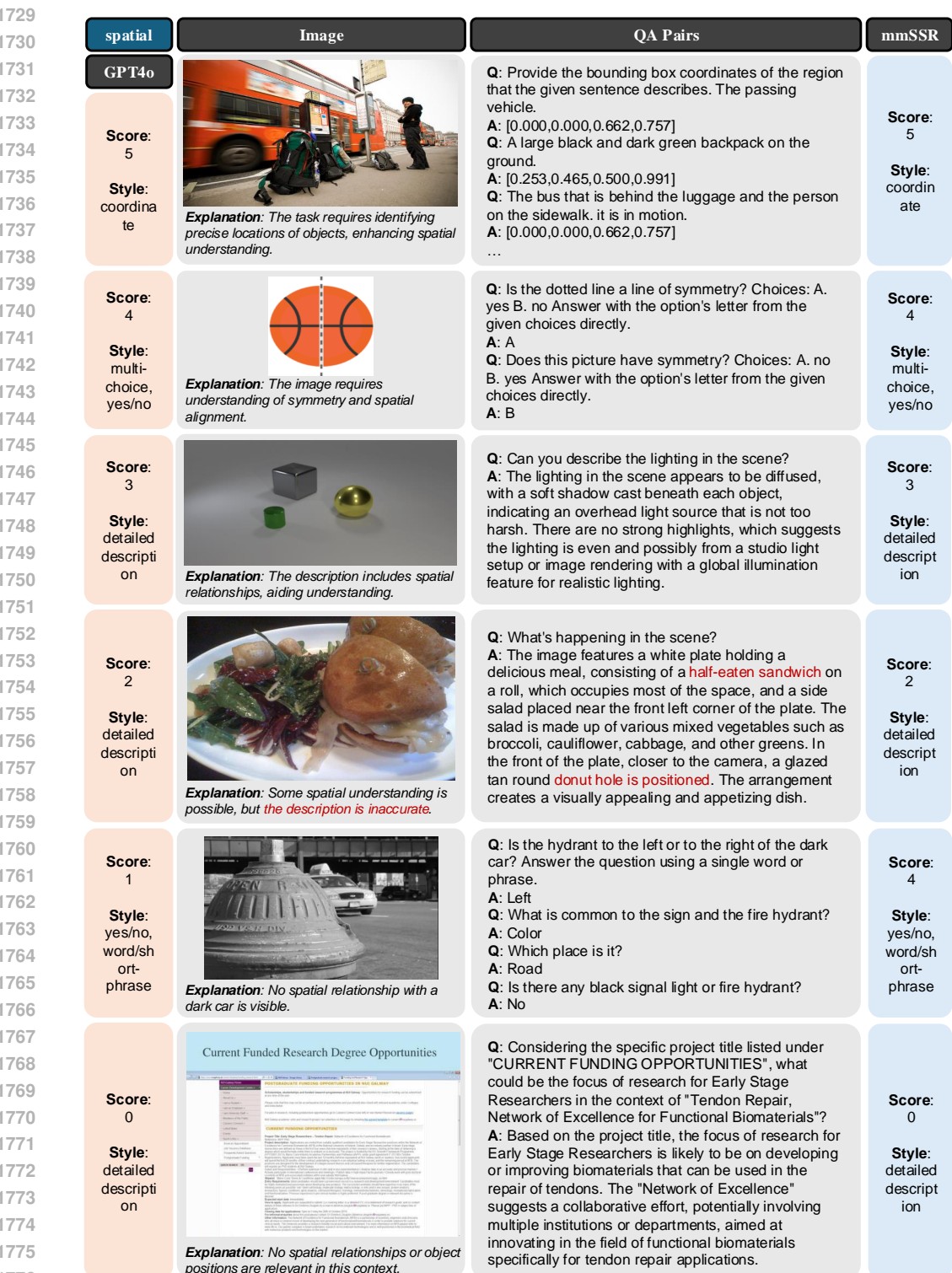

Figure 20: Randomly selected samples from the LLaVA-OVSI data pool within each score group for the **object spatial understanding** capability, ranging from 5 to 0, as evaluated by GPT-4o, along with corresponding score explanations and mmSSR predictions.

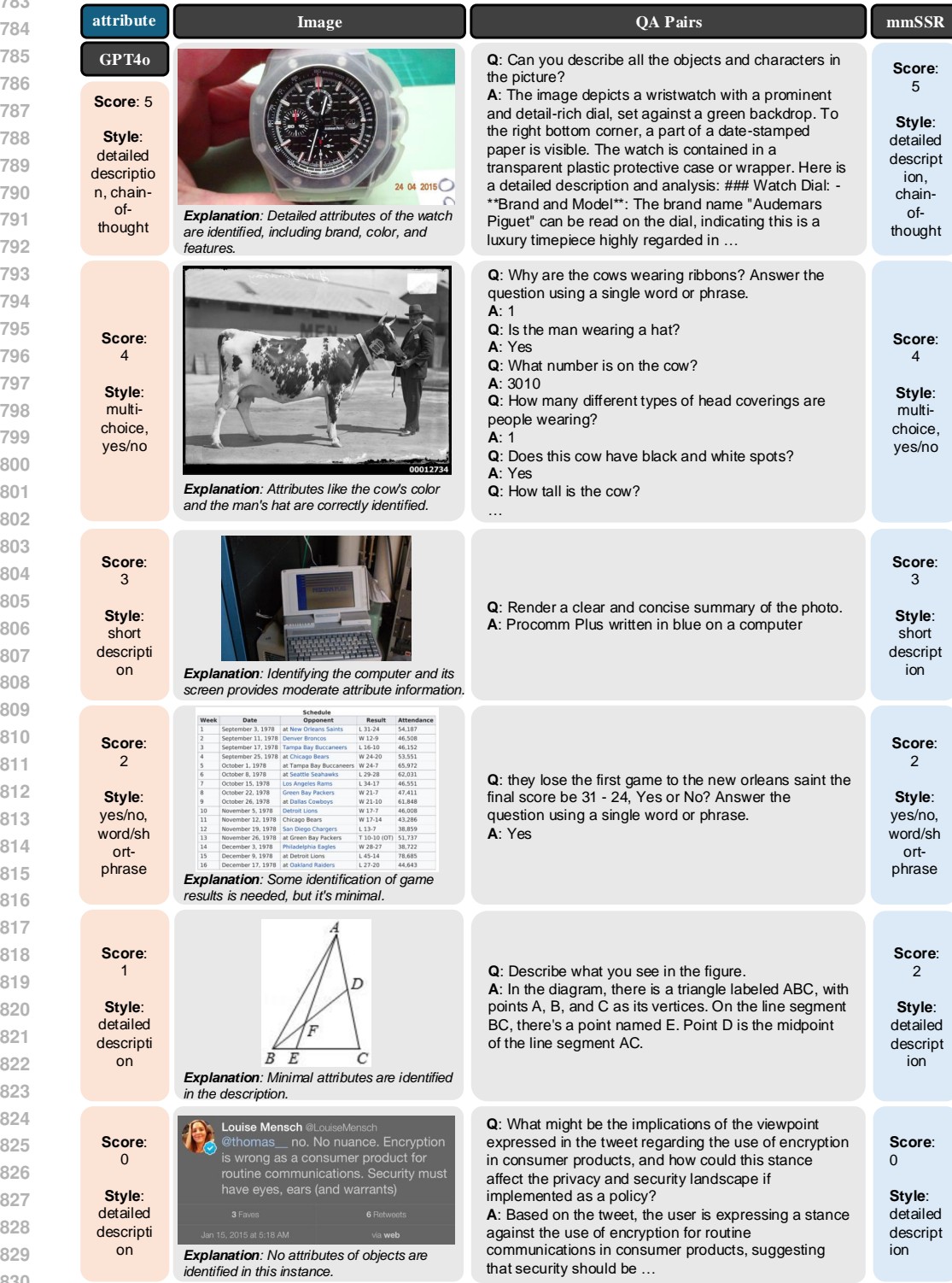

Figure 21: Randomly selected samples from the LLaVA-OVSI data pool within each score group for the **attribute identification** capability, ranging from 5 to 0, as evaluated by GPT-4o, along with corresponding score explanations and mmSSR predictions.

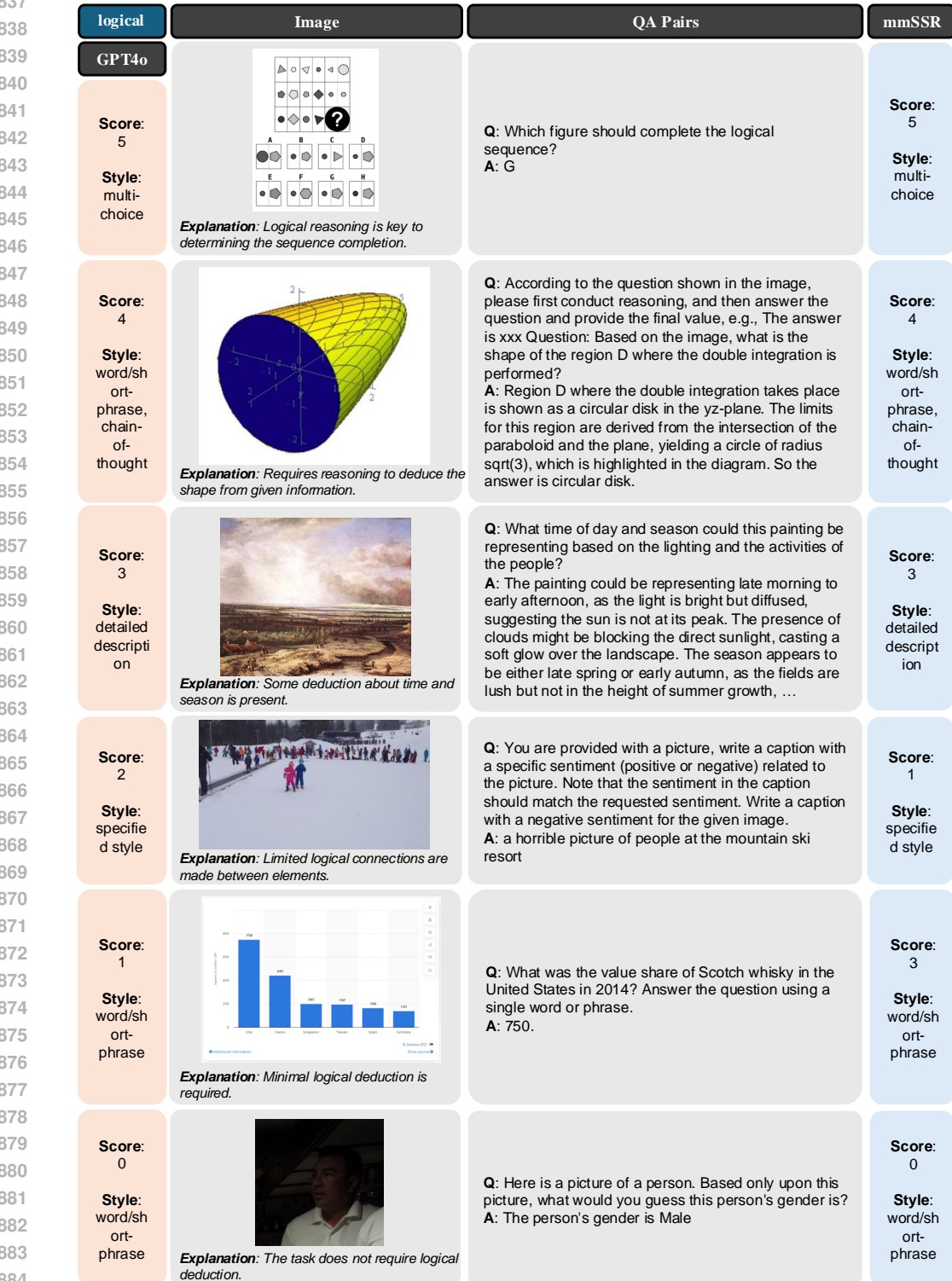

| logical | Image | QA Pairs | mmSSR |
|---|---|---|---|
| **GPT4o** | | | |
| **Score**: 5 **Style**: multi-choice | *Explanation: Logical reasoning is key to determining the sequence completion.* | **Q**: Which figure should complete the logical sequence? **A**: G | **Score**: 5 **Style**: multi-choice |
| **Score**: 4 **Style**: word/short-phrase, chain-of-thought | *Explanation: Requires reasoning to deduce the shape from given information.* | **Q**: According to the question shown in the image, please first conduct reasoning, and then answer the question and provide the final value, e.g., The answer is xxx Question: Based on the image, what is the shape of the region D where the double integration is performed? **A**: Region D where the double integration takes place is shown as a circular disk in the yz-plane. The limits for this region are derived from the intersection of the paraboloid and the plane, yielding a circle of radius sqrt(3), which is highlighted in the diagram. So the answer is circular disk. | **Score**: 4 **Style**: word/short-phrase, chain-of-thought |
| **Score**: 3 **Style**: detailed description | *Explanation: Some deduction about time and season is present.* | **Q**: What time of day and season could this painting be representing based on the lighting and the activities of the people? **A**: The painting could be representing late morning to early afternoon, as the light is bright but diffused, suggesting the sun is not at its peak. The presence of clouds might be blocking the direct sunlight, casting a soft glow over the landscape. The season appears to be either late spring or early autumn, as the fields are lush but not in the height of summer growth, … | **Score**: 3 **Style**: detailed description |
| **Score**: 2 **Style**: specified style | *Explanation: Limited logical connections are made between elements.* | **Q**: You are provided with a picture, write a caption with a specific sentiment (positive or negative) related to the picture. Note that the sentiment in the caption should match the requested sentiment. Write a caption with a negative sentiment for the given image. **A**: a horrible picture of people at the mountain ski resort | **Score**: 1 **Style**: specified style |
| **Score**: 1 **Style**: word/short-phrase | *Explanation: Minimal logical deduction is required.* | **Q**: What was the value share of Scotch whisky in the United States in 2014? Answer the question using a single word or phrase. **A**: 750. | **Score**: 3 **Style**: word/short-phrase |
| **Score**: 0 **Style**: word/short-phrase | *Explanation: The task does not require logical deduction.* | **Q**: Here is a picture of a person. Based only upon this picture, what would you guess this person's gender is? **A**: The person's gender is Male | **Score**: 0 **Style**: word/short-phrase |

Figure 22: Randomly selected samples from the LLaVA-OVSI data pool within each score group for the **logical deduction** capability, ranging from 5 to 0, as evaluated by GPT-4o, along with corresponding score explanations and mmSSR predictions.

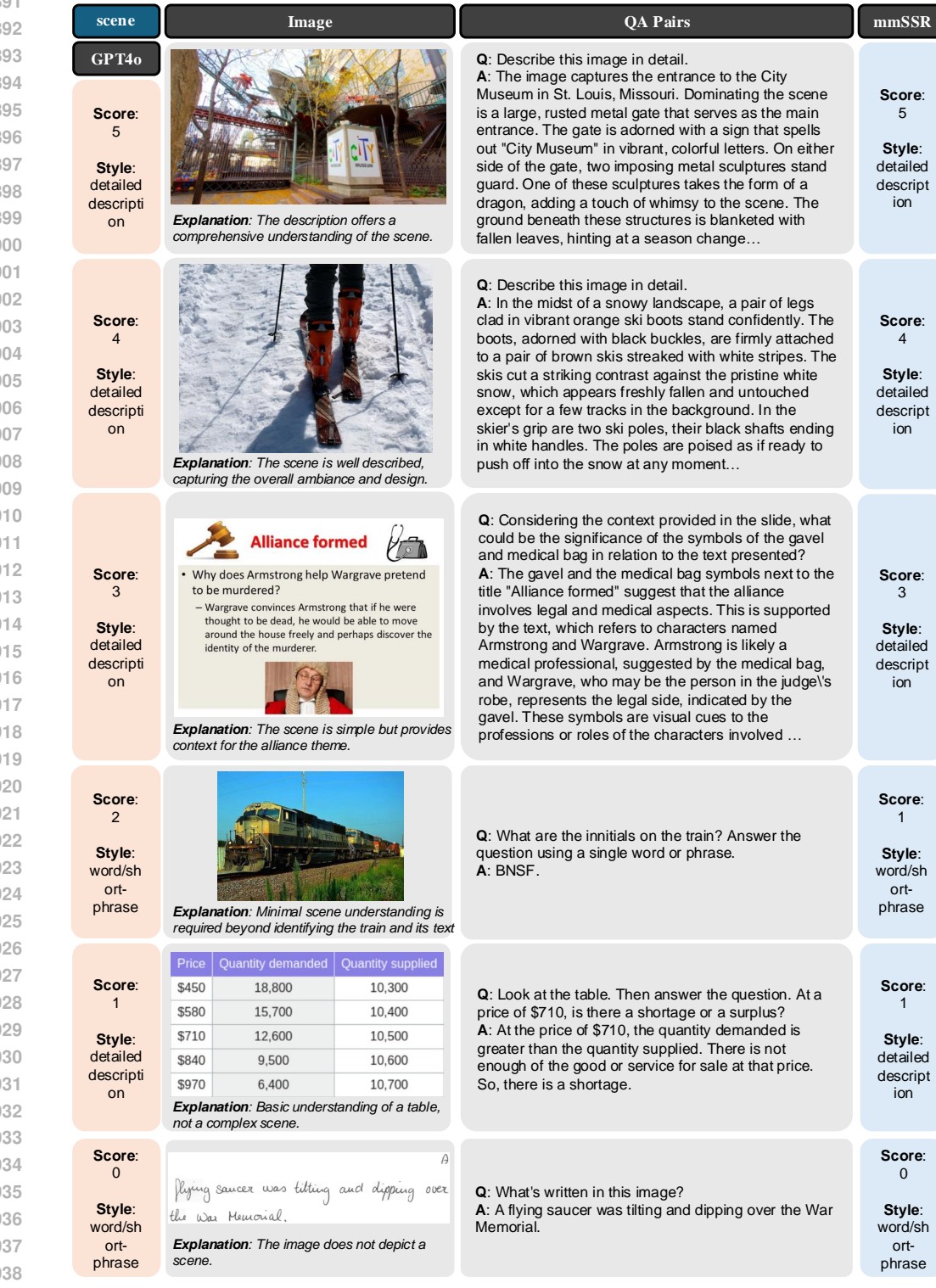

| scene | Image | QA Pairs | mmSSR |
|---|---|---|---|
| **GPT4o** | *Explanation: The description offers a comprehensive understanding of the scene.* | **Q**: Describe this image in detail.
**A**: The image captures the entrance to the City Museum in St. Louis, Missouri. Dominating the scene is a large, rusted metal gate that serves as the main entrance. The gate is adorned with a sign that spells out "City Museum" in vibrant, colorful letters. On either side of the gate, two imposing metal sculptures stand guard. One of these sculptures takes the form of a dragon, adding a touch of whimsy to the scene. The ground beneath these structures is blanketed with fallen leaves, hinting at a season change… | **Score**: 5

**Style**: detailed description |
| **Score**: 5

**Style**: detailed description | | | |
| **Score**: 4

**Style**: detailed description | *Explanation: The scene is well described, capturing the overall ambiance and design.* | **Q**: Describe this image in detail.
**A**: In the midst of a snowy landscape, a pair of legs clad in vibrant orange ski boots stand confidently. The boots, adorned with black buckles, are firmly attached to a pair of brown skis streaked with white stripes. The skis cut a striking contrast against the pristine white snow, which appears freshly fallen and untouched except for a few tracks in the background. In the skier's grip are two ski poles, their black shafts ending in white handles. The poles are poised as if ready to push off into the snow at any moment… | **Score**: 4

**Style**: detailed description |
| **Score**: 3

**Style**: detailed description | *Explanation: The scene is simple but provides context for the alliance theme.* | **Q**: Considering the context provided in the slide, what could be the significance of the symbols of the gavel and medical bag in relation to the text presented?
**A**: The gavel and the medical bag symbols next to the title "Alliance formed" suggest that the alliance involves legal and medical aspects. This is supported by the text, which refers to characters named Armstrong and Wargrave. Armstrong is likely a medical professional, suggested by the medical bag, and Wargrave, who may be the person in the judge\'s robe, represents the legal side, indicated by the gavel. These symbols are visual cues to the professions or roles of the characters involved … | **Score**: 3

**Style**: detailed description |
| **Score**: 2

**Style**: word/short-phrase | *Explanation: Minimal scene understanding is required beyond identifying the train and its text* | **Q**: What are the innitials on the train? Answer the question using a single word or phrase.
**A**: BNSF. | **Score**: 1

**Style**: word/short-phrase |
| **Score**: 1

**Style**: detailed description | *Explanation: Basic understanding of a table, not a complex scene.* | **Q**: Look at the table. Then answer the question. At a price of $710, is there a shortage or a surplus?
**A**: At the price of $710, the quantity demanded is greater than the quantity supplied. There is not enough of the good or service for sale at that price. So, there is a shortage. | **Score**: 1

**Style**: detailed description |
| **Score**: 0

**Style**: word/short-phrase | *Explanation: The image does not depict a scene.* | **Q**: What's written in this image?
**A**: A flying saucer was tilting and dipping over the War Memorial. | **Score**: 0

**Style**: word/short-phrase |

Figure 23: Randomly selected samples from the LLaVA-OVSI data pool within each score group for the **scene understanding** capability, ranging from 5 to 0, as evaluated by GPT-4o, along with corresponding score explanations and mmSSR predictions.

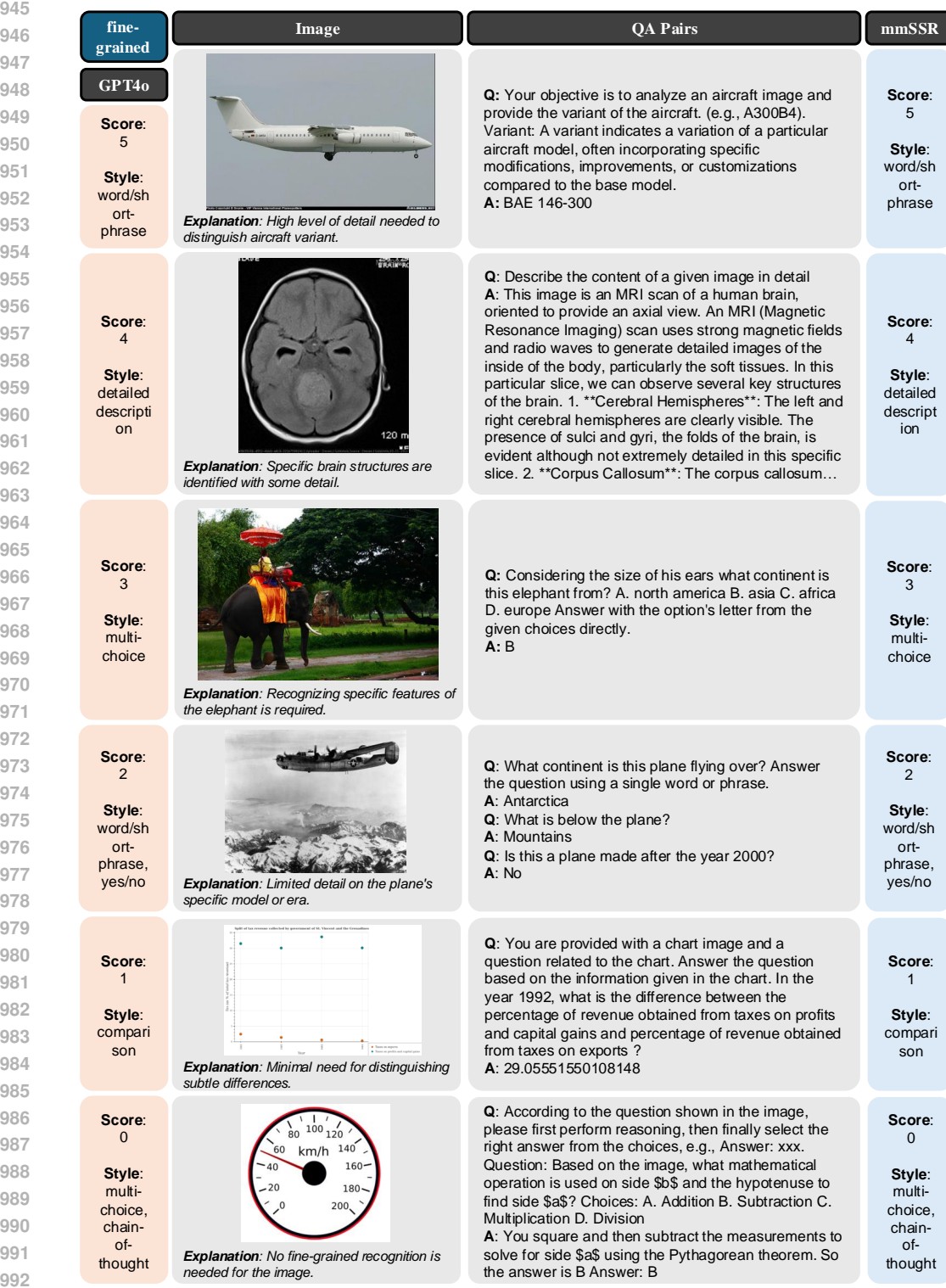

Figure 24: Randomly selected samples from the LLaVA-OVSI data pool within each score group for the **fine-grained recognition** capability, ranging from 5 to 0, as evaluated by GPT-4o, along with corresponding score explanations and mmSSR predictions.

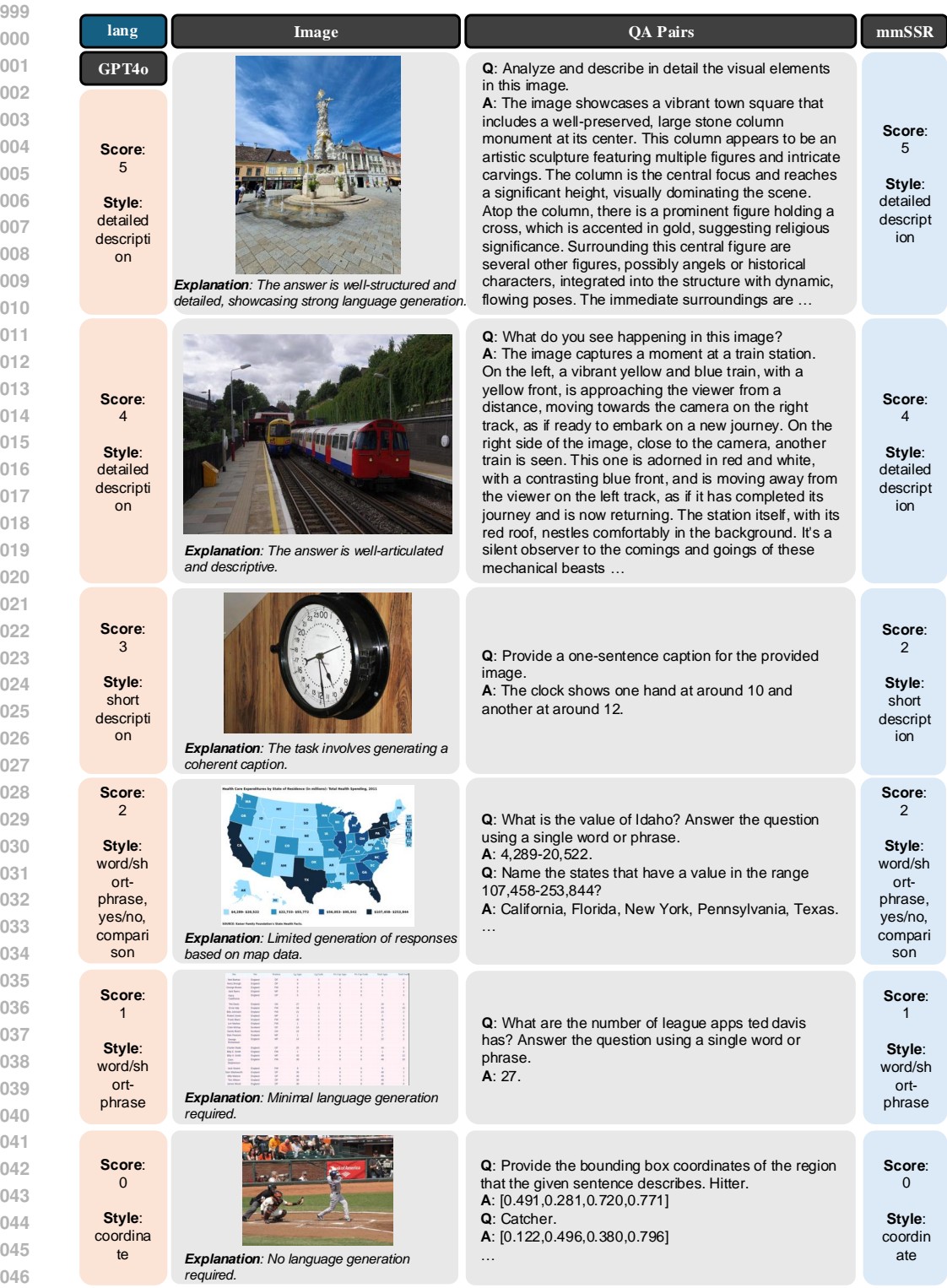

Figure 25: Randomly selected samples from the LLaVA-OVSI data pool within each score group for the **language generation** capability, ranging from 5 to 0, as evaluated by GPT-4o, along with corresponding score explanations and mmSSR predictions.

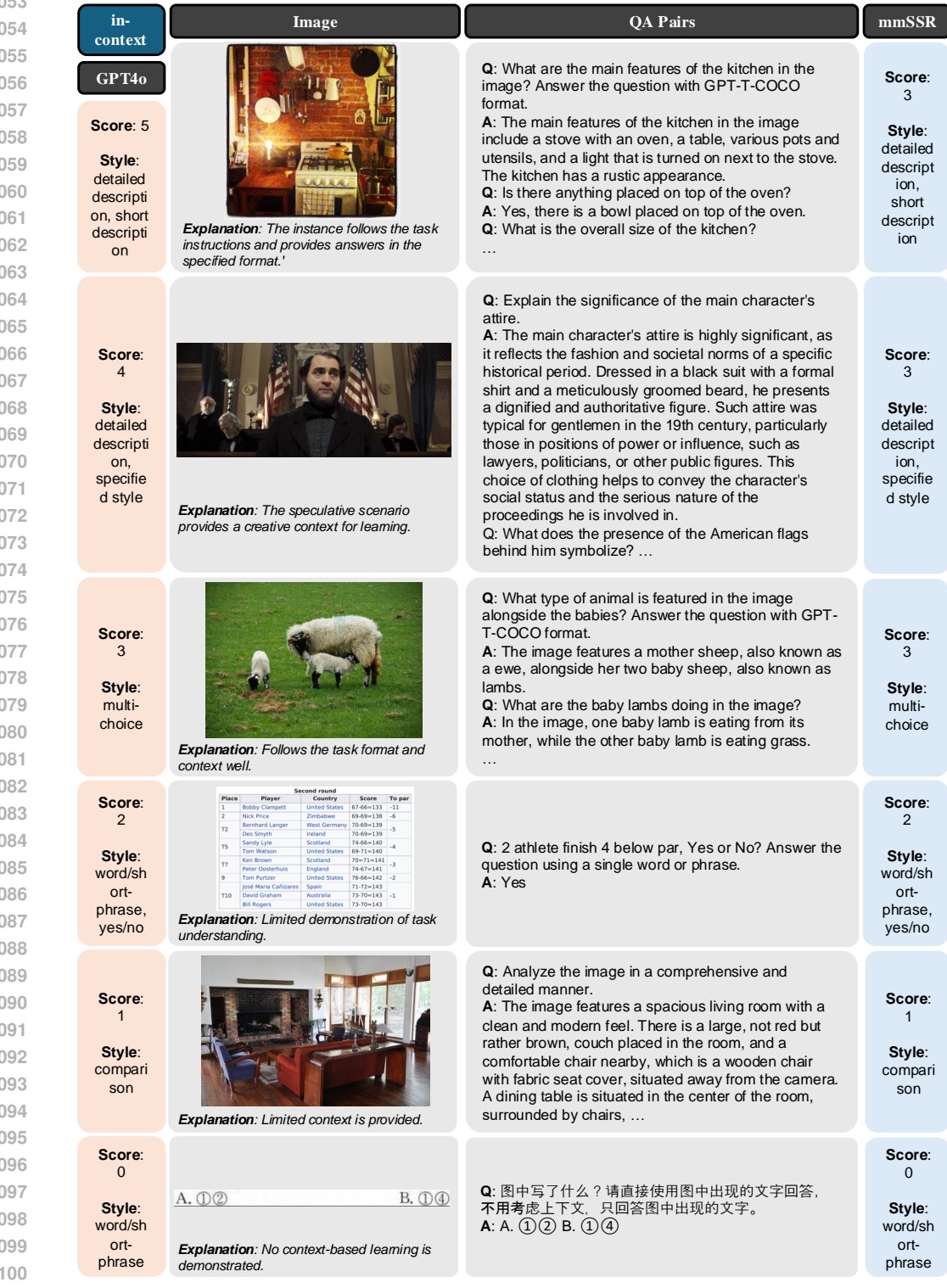

Figure 26: Randomly selected samples from the LLaVA-OVSI data pool within each score group for the **in-context learning** capability, ranging from 5 to 0, as evaluated by GPT-4o, along with corresponding score explanations and mmSSR predictions.

| OCR | Image | QA Pairs | mmSSR |
|---|---|---|---|
| **GPT4o** | | | |
| **Score**: 5 

 **Style**: detailed description |  *Explanation*: The instance provides a clear example of handwritten text recognition, which is highly relevant for OCR capability. | **Q**: What is scribbled in this image? 
 **A**: For the most part therefore local employment can be provided only by the expansion of existing industry and the introduction of new factories, laboratories and offices, and the expected but not easily stimulated development of the service industries. | **Score**: 5 

 **Style**: detailed description |
| **Score**: 4 

 **Style**: detailed description |  *Explanation*: The description includes details about the text on the card, aiding OCR capabilities. | **Q**: Describe this image in detail. 
 **A**: The image displays a golden business card with intricate, laser-cut detailing around the edges. The card features a mix of English and Chinese text. In the upper portion, the Chinese characters are prominent, and below them, the English translation reads "Personal Tailor." The text is printed in red with a shimmering effect that contrasts with the metallic sheen of the gold card. The card's design is elegant and suggests a personalized service, … | **Score**: 4 

 **Style**: detailed description |
| **Score**: 3 

 **Style**: short description |  *Explanation*: The image contains text on the truck, which can be recognized. | **Q**: Write a terse but informative summary of the picture. 
 **A**: The yellow taxi drives next to a freshdirect truck. | **Score**: 3 

 **Style**: short description |
| **Score**: 2 

 **Style**: short description |  *Explanation*: Some text is visible on the devices but not the main focus.' | **Q**: Provide a one-sentence caption for the provided image. 
 **A**: Five different types of cellular phones next to each other on a table. | **Score**: 2 

 **Style**: short description |
| **Score**: 1 

 **Style**: detailed description |  *Explanation*: Minimal relevance as there is text in the image but not utilized. | **Q**: Considering the presence of multiple bottles in the scene, what issue does this environment illustrate, and how can it affect pedestrians? Answer the question with GPT-T-COCO format. 
 **A**: The environment in the image shows an issue of littering, as multiple bottles are scattered across the scene near the road. This can negatively affect pedestrians in several ways, such as creating a dirty and unkempt surrounding, posing health risks, … | **Score**: 0 

 **Style**: detailed description |
| **Score**: 0 

 **Style**: multi-choice |  *Explanation*: No text is present in the image.' | **Q**: You will be given an image of a guitar. The image could be of different categories like painting, cartoon, photograph, or sketch. Identify the image category. Options: (a) Photograph (b) Cartoon (c) Art painting (d) Sketch 
 **A**: (d) Sketch | **Score**: 0 

 **Style**: multi-choice |

Figure 27: Randomly selected samples from the LLaVA-OVSI data pool within each score group for the **optical character recognition** capability, ranging from 5 to 0, as evaluated by GPT-4o, along with corresponding score explanations and mmSSR predictions.

# F  THE USE OF LARGE LANGUAGE MODELS

**Role in Experimental Validation.**  GPT-4o, Qwen2-VL-7B, and LLaVA-OVSI-7B were used to generate capability scores and interactive styles for instructional data. We first curated a small corpus of data annotated by the proprietary GPT-4o to fine-tune scorer and styler models, which can be applied directly to new datasets, as validated in Sec. 4.3. Concurrently, we employed open-source alternatives for scoring and styling to validate that mmSSR is robust and orthogonal to the choice of MLLM, as demonstrated by the analysis in Sec. 4.2.

**Role in Writing and Editing.**  We also used GPT-4o and Gemini-2.5-Pro to polish the language of the manuscript, including grammar correction and clarity improvement.

We supervised this process to ensure its accuracy and originality, and we take full responsibility for the final content of this paper.