# OpenReview forum: "mmSSR: Harvesting Rich, Scalable and Transferable Multi-Modal Data for Instruction Fine-Tuning"
_ICLR.cc/2026/Conference — ICLR 2026 Conference Withdrawn Submission_

### Official Review · Reviewer_kDC1 · 2025-10-27

**Soundness:** 1
**Presentation:** 3
**Contribution:** 2
**Rating:** 2
**Confidence:** 5

**Summary:**

This paper introduces mmSSR, a scalable method for selecting high-quality, diverse, and transferable data from massive multi-modal (vision-language) datasets for instruction fine-tuning.

Instead of relying on computationally expensive embedding-based clustering or vague quality metrics, the method proposes a new evaluation framework:

Multi-modal Rich Scorers: Decomposes data "quality" into 14+ specific, interpretable vision-language capabilities (e.g., spatial understanding, logical deduction, STEM knowledge).

Multi-modal Rich Styler: Decomposes "diversity" into 9+ interaction styles (e.g., multi-choice, detailed description, chain-of-thought) as a more efficient proxy for diversity.

The pipeline works by first using a powerful "super model" (like GPT-4o) to annotate a small fraction of the data with these capability scores and style labels. Then, a smaller, efficient proxy model (the mmSSR) is fine-tuned on these annotations. This proxy model is then used to rapidly score and classify the entire large-scale dataset. Finally, a "style-aware, score-prioritized" sampling strategy (Round-Robin) is used to select a small, balanced, high-quality subset.

The authors demonstrate that using only 30% of a 2.6 million-sample dataset selected by mmSSR achieves 99.1% of the performance of training on the full dataset, proving the method's efficiency and scalability.

**Strengths:**

This approach, combined with the "style-aware, score-prioritized" sampling, achieves remarkable data efficiency, evidenced by the main result: achieving 99.1% of full performance using only 30% of the 2.6 million-sample dataset.

**Weaknesses:**

Major Weaknesses

1. Heavy Reliance on Predefined Metrics and "Super Models"

The paper's methodology is heavily dependent on the "Rich Scorers" and "Rich Styler," which rely on a set of manually predefined factors (14 capabilities and 9 styles). This raises several significant concerns:

Generalizability and Completeness: How can the generalizability of these manually defined metrics be guaranteed? How do the authors justify that this specific set of factors is both reasonable and comprehensive for evaluating multi-modal instructional data? The paper lacks a strong argument for why these 14+9 factors are the right ones.

Alignment with the "Super Model": The method requires a "super model" (e.g., GPT-4o) to generate the initial labels. What is the alignment degree between the authors' definitions of these factors and the model's interpretation? How can the authors ensure that the "super model" accurately understands their intent and expresses the scores reliably?

Reliability and Cost: Even if a "super model" is available, its reliability is a concern. Furthermore, what is the overall cost (computational and/or financial) of using this model for annotation? Are there potential data leakage risks?

Practicality: The method's dependency on a powerful, proprietary "super model" is a critical bottleneck. What is the alternative if such a model is unavailable? This scenario represents a significant challenge for real-world data selection, and the paper's solution seems impractical outside of a resource-rich environment.

2. Lack of Multi-modal Specificity

The paper does not adequately explain how the proposed method is specifically tailored for multi-modal data. The defined capabilities (e.g., "logical deduction") and styles (e.g., "chain-of-thought") seem largely applicable to text-only scenarios. The methodology, as described, appears to be almost entirely transferable to selecting data for text-only LLMs without significant modification. The unique challenges and characteristics of multi-modal (vision-language) data selection are not sufficiently addressed.

3. Missing Discussion of Recent State-of-the-Art Work

The related work section is notably outdated. It almost exclusively discusses methods from 2024 or earlier. Given the rapid development in this research area, the omission of more recent and state-of-the-art (SOTA) work is a significant oversight. While reproducing all recent methods is time-consuming, completely ignoring them in the discussion is unacceptable and significantly diminishes the paper's academic value and contribution.

The authors should, at a minimum, discuss and contextualize their work against more recent methods, such as:

[1] OASIS: Online Sample Selection for Continual Visual Instruction Tuning

[2] Prism: Self-pruning intrinsic selection method for training-free multimodal data selection

[3] MLLM-Selector: Necessity and Diversity-driven High-Value Data Selection for Enhanced Visual Instruction Tuning

Minor Weaknesses

Inconsistent Terminology: The terminology throughout the paper is confusing. The terms "MLLM" and "VLM" are used interchangeably and inconsistently. The authors should unify their terminology for clarity.

Limited Evaluation Benchmarks: The evaluation should be expanded to include other important benchmarks. For instance, adding a test like POPE would provide a more comprehensive assessment, particularly regarding the model's tendency for object hallucination.

**Questions:**

See Above.

---

### Official Review · Reviewer_6tmZ · 2025-10-28

**Soundness:** 3
**Presentation:** 3
**Contribution:** 2
**Rating:** 4
**Confidence:** 3

**Summary:**

This paper introduces mmSSR, a scalable multi-modal data selection framework for fine-tuning large multimodal language models (MLLMs). Instead of relying on random or embedding-based sampling, mmSSR decomposes quality evaluation into interpretable vision-language metrics and uses multi-modal scorers and stylers to rank and partition data by quality and diversity. The method efficiently scales to millions of samples, supports capability-specific and cross-domain data curation, and improves stability and generalization.

**Strengths:**

* The paper makes a meaningful contribution by identifying the limitations of using abstract scores in data selection for large multimodal models and instead advocates for fine-grained evaluation across multiple vision-language capabilities and styles. This shift enables a more interpretable and precise understanding of data quality.

* The proposed mmSSR framework is validated through extensive experiments that convincingly demonstrate its effectiveness, outperforming baseline methods and state-of-the-art selection strategies across multiple multimodal benchmarks. These results highlight both the scalability and robustness of the approach.

* The paper is clearly written and well-organized, with a logical flow from motivation to methodology and experiments. The clarity of explanation and structure make the technical content accessible and easy to follow.

**Weaknesses:**

* The proposed idea does not appear particularly novel. The observed performance improvements seem to primarily result from more fine-grained prompt engineering, which provides better instructions to a powerful VLM for data selection, followed by distilling this large model through fine-tuning a smaller model on the selected data. However, the general concept of leveraging LLM for data selection has already been widely explored. In this context, the use of fine-grained prompts appears to be an incremental and relatively straightforward extension rather than a fundamentally new idea.

* Since mmSSR depends on powerful models such as GPT-4o to perform data selection, it would strengthen the empirical evaluation to include comparisons against existing baselines that also employ LLMs for this purpose. This would provide a fairer assessment of the method’s true contribution beyond differences in underlying model capacity.

**Questions:**

Is it possible that selecting more data on some subcategories or styles can lead to better performance?

---

### Official Review · Reviewer_d3qf · 2025-10-31

**Soundness:** 3
**Presentation:** 3
**Contribution:** 3
**Rating:** 6
**Confidence:** 3

**Summary:**

This paper presents MMSSR, a structured framework for scalable and transferable multi-modal data selection in instruction fine-tuning. Instead of relying on a single opaque quality score, the approach decomposes data quality into two interpretable components: a multi-capability space that reflects semantic and reasoning abilities, and a style space that captures linguistic and interactional diversity. Within this formulation, data sampling is carried out through a style-aware, capability-prioritized selection mechanism designed to produce balanced and informative subsets efficiently.

The framework comprises three modules. Rich Scorers use a capability predictor trained on GPT-4o annotations to represent each sample as a fourteen-dimensional vector covering interpretable vision–language abilities such as spatial reasoning, OCR recognition, logical deduction, and STEM-related reasoning. Rich Styler introduces a lightweight classifier that identifies nine interaction styles—ranging from task-oriented to explanatory or conversational—providing a semantic measure of stylistic diversity beyond embedding-based similarity. The Round-Robin Selector performs alternating sampling across style groups while prioritizing higher-scored samples within each, aiming to balance sample quality and stylistic coverage.

Empirical evaluation is conducted on the LLaVA-OneVision single-image instruction-tuning stage, using fourteen multimodal benchmarks including MMBench, MMStar, ScienceQA, and MathVistaMini. The reported results indicate that MMSSR achieves a level of performance close to full-data training (approximately 99 %) while using only 30 % of the data, and shows consistent improvements over prior data selection methods such as Deita, COINCIDE, and ICONS. Ablation analyses further suggest that both capability richness and style modeling contribute to the observed performance and stability.

**Strengths:**

- The paper addresses the problem of multi-modal data selection through a structured framework that models both capability and interaction style, offering a clear formulation of the task.
- The proposed selection process is interpretable, as each sample is characterized by a capability vector and a style label, allowing analysis of the factors influencing sample choice.
- The experimental setup covers 14 multimodal benchmarks, and the reported results indicate consistent improvements over several existing data selection methods under different sampling budgets.
- The scorers and stylers are evaluated across datasets and model architectures, showing reasonable transfer performance.
- The paper is generally well structured and the presentation of the methodology and results is clear.

**Weaknesses:**

- The definition of the 14 capability dimensions is manually designed. The paper does not provide evidence that these dimensions are optimal or exhaustive, and it is unclear how well they generalize to tasks involving other modalities.
- The modeling of interaction styles is discrete, assuming nine separate categories. The framework does not explore whether style variation may be continuous or overlapping.
- The approach depends on a large teacher model (GPT-4o) for initial capability and style annotation, which could introduce non-trivial resource requirements for replication.
- The ablation analysis verifies the importance of each module but does not isolate the contribution of individual capability dimensions, limiting insight into how specific abilities affect performance.
- The sampling mechanism allocates data across style categories using fixed proportions. The potential effects of this heuristic on the overall sample distribution are not fully discussed.

**Questions:**

see weakness

---

### Official Review · Reviewer_bZ2f · 2025-11-02

**Soundness:** 2
**Presentation:** 2
**Contribution:** 2
**Rating:** 4
**Confidence:** 2

**Summary:**

The paper proposes mmSSR (Multi-Modal Rich Scorer and Styler), a novel data selection framework designed for multi-modal large language models during instruction fine-tuning. Recognizing that previous data selection strategies (like LIMA and Deita) for LLMs often fail to generalize to multi-modal data, the authors decompose the notion of “data quality” into interpretable vision-language capabilities (e.g., spatial reasoning, logical deduction, OCR, etc.) and interaction styles (e.g., conversational tone or instruction form).

**Strengths:**

Novel and interpretable data selection paradigm

By decomposing abstract data quality into interpretable, model-attributable capabilities, mmSSR provides a transparent and controllable alternative to embedding-based or black-box selection approaches.

The use of style clustering as a lightweight diversity mechanism avoids computationally expensive embedding similarity calculations.

**Weaknesses:**

1. Complexity and annotation cost: The multi-capability scoring (14 criteria) and style identification pipelines introduce non-trivial overhead in data labeling and model finetuning, which could challenge scalability in low-resource environments.

2. Lack of qualitative analysis on selected data: The paper convincingly shows quantitative improvements but provides limited qualitative visualization of the differences between mmSSR-selected and random samples (e.g., examples of diverse styles or capability distributions).

**Questions:**

Please refer to the weakness part.

---

### Note · Authors · 2025-11-14

I have read and agree with the venue's withdrawal policy on behalf of myself and my co-authors.